# FOLLOW-THE-PERTURBED-LEADER FOR DECOUPLED BANDITS: BEST-OF-BOTH-WORLDS AND PRACTICALITY

## ABSTRACT

We study the decoupled multi-armed bandit (MAB) problem, where the learner selects one arm for exploration and one arm for exploitation in each round. The loss of the explored arm is observed but not counted, while the loss of the exploited arm is incurred without being observed. We propose a policy within the Follow-the-Perturbed-Leader (FTPL) framework using Pareto perturbations. Our policy achieves (near-)optimal regret regardless of the environment, i.e., Best-of-Both-Worlds (BOBW): constant regret in the stochastic regime, improving upon the optimal bound of the standard MABs, and minimax optimal regret in the adversarial regime. Moreover, the practicality of our policy stems from avoiding both the convex optimization step required by the previous BOBW policy, Decoupled-Tsallis-INF (Rouyer & Seldin, 2020), and the resampling step that is typically necessary in FTPL. Consequently, it achieves substantial computational improvement, about 20 times faster than Decoupled-Tsallis-INF, while also demonstrating better empirical performance in both regimes. Finally, we empirically show that our approach outperforms a pure exploration policy, and that naively combining a pure exploration with a standard exploitation policy is suboptimal.

## 1 INTRODUCTION

The multi-armed bandit (MAB) is a fundamental framework in sequential decision-making, widely applied in areas such as recommender systems (Brodén et al., 2017; Zhou et al., 2017), dynamic pricing (Misra et al., 2019; Mueller et al., 2019), and sequential experimental design (Burtini et al., 2015). Imagine a scenario in the standard MAB setting: selecting among $K$ arms over a time horizon $T$. The learner aims to minimize the cumulative regret, the difference between the total loss actually incurred and that of the best fixed arm in hindsight. Since only the loss $\ell_{t,i_t}$ of the selected arm $i_t \in [K]$ is observed at each round $t \in [T]$, the learner must balance exploiting promising arms with exploring seemingly suboptimal ones to gather information. In other words, a single decision is made under the consideration of both objectives, meaning that exploitation and exploration are *coupled* within each round.

Although this formulation covers many practical applications, it does not capture scenarios where exploration can be performed independently of exploitation. For instance, in ultra-wide band (UWB) communication systems, the learner can sense a channel different from currently used for transmission, to observe feedback and avoid mutual interference (Avner et al., 2012). Moreover, when a real-time system operates alongside a high-fidelity simulator, the learner can explore in the simulator and exploit in the real system, such as sim-to-real transfer in robotics (Zhao et al., 2020) without degrading real-world performance. A related example arises in recommender systems (Che et al., 2025), where the platform, given user context (e.g., preferences), can explore with a random subset of users to update the policy, while exploiting the rest by serving the best-known items.

To model such scenarios, Avner et al. (2012) introduced the *decoupled* MAB setting, where the learner can select two arms at each round: one for observing the loss without incurring it, and one for incurring the loss without observing it. This decoupling of exploration and exploitation recovers the standard MAB, when the learner is restricted to select same arm for both objectives. Note that this framework differs from pure exploration problems, in which the primary focus is on exploration

to identify the best (good) arm (Even-Dar et al., 2006). It is also different from explore-then-commit (ETC) style policies, which divide exploration and exploitation into distinct phases, performing only one of the two at each round (Garivier et al., 2016).

In the decoupled adversarial MAB, Avner et al. (2012) established a lower bound of $\Omega(\sqrt{KT})$, matching that of the standard MAB (Auer et al., 2002), indicating that the two problems are similarly challenging. Against an oblivious adversary, their Exp3-type policy obtained an adversarial regret of $\mathcal{O}(\sqrt{KT \ln K})$, which improves to $\mathcal{O}(\sqrt{T \ln K})$ when a single dominant arm exists. Similarly, they achieved a regret of $\mathcal{O}(\sqrt{T \ln K})$ in the decoupled stochastic MAB with a unique optimal arm. However, this is highly suboptimal, as even an anytime sampling rule designed for pure exploration tasks attains a time-independent cumulative regret of $\tilde{\mathcal{O}}(K^3/\Delta_{\min}^2)$ in the same setting, despite being aimed at minimizing the expected simple regret (Jourdan et al., 2023). Here, $\Delta_{\min} = \min_{i:\Delta_i>0} \Delta_i$ denotes the minimum suboptimality gap, where $\Delta_i = \mathbb{E}[\ell_{\cdot,i}] - \min_j \mathbb{E}[\ell_{\cdot,j}]$.

In addition to these limitations, the update rules for arm-selection probabilities and choice of learning rates proposed by Avner et al. (2012) require prior knowledge of both the time horizon and the environment. In practice, however, the nature of the environment is typically unknown, which motivates the design of policies that guarantee (near-)optimal performance across all possible environments, known as the Best-of-Both-Worlds (BOBW) guarantee (Bubeck & Slivkins, 2012).

Tsallis-INF, based on the Follow-the-Regularized-Leader (FTRL) framework, is a prominent BOBW policy for standard MABs (Zimmert & Seldin, 2021). Extending this to the decoupled setting, Rouyer & Seldin (2020) proposed Decoupled-Tsallis-INF, which also achieves BOBW: minimax optimal $\mathcal{O}(\sqrt{KT})$ regret in the adversarial regime and near-optimal time-independent regret of $\mathcal{O}(K/\Delta_{\min})$ in the stochastic regime. This result marks a significant improvement over Avner et al. (2012) and outperforms the optimal bound for the standard stochastic MAB, $\mathcal{O}(\sum_{i:\Delta_i>0} \log T/\Delta_i)$.

Despite its strong theoretical guarantees, a practical drawback of the FTRL framework is the need to solve a convex optimization problem at every round to compute arm-selection probabilities, which can be computationally intensive. As a more efficient alternative, the Follow-the-Perturbed-Leader (FTPL) framework, of which Exp3 is a special case (Auer et al., 2002; Bubeck & Cesa-Bianchi, 2012), has been studied in standard MABs and achieves BOBW without requiring any convex optimization (Honda et al., 2023; Lee et al., 2024). In the decoupled MAB setting, however, only the suboptimal result is known for FTPL-type policy (Avner et al., 2012). Hence, the following research question arise: *Can we achieve BOBW for decoupled bandits while improving regret performance and computational efficiency?*

## 1.1 CONTRIBUTION

A key computational challenge in FTRL and FTPL bandit policies lies in computing arm-selection probabilities, which are required (i) to select an arm and (ii) to construct an unbiased loss estimator, typically using an importance-weighted (IW) estimator. In FTRL, these probabilities are obtained by solving a convex optimization problem in each round. By contrast, FTPL avoids this by selecting arms through random perturbations, without explicitly computing probabilities. This is sufficient for (i), but creates a difficulty for (ii), since the IW estimator requires the probability of the selected arm. To estimate this, FTPL typically relies on geometric resampling (Neu & Bartók, 2016) or its efficient variant (Chen et al., 2025), which incur a per-step cost of $\mathcal{O}(K^2)$ or $\mathcal{O}(K \log K)$, respectively. The latter is less costly than the convex optimization step of Tsallis-INF in standard MABs. However, both methods cannot be directly extended to the decoupled setting. The reason is that resampling methods only estimate the probability of the selected arm, whereas the decoupled setting requires the full exploitation probability vector to compute exploration probability vector.

In this paper, we propose a decoupled FTPL policy that achieves BOBW *without convex optimization or resampling*, attaining the same regret order as BOBW FTRL policy while substantially reducing computational cost. Our main contributions are as follows:

- We introduce a simple FTPL policy (Algorithm 1) with Pareto perturbations for decoupled MAB, the first use of Pareto perturbations in this setting.

- We achieve $\mathcal{O}(\sqrt{KT})$ regret in the adversarial regime (Theorem 1) and $\mathcal{O}(K/\Delta_{\min})$ regret in the stochastic regime (Theorem 2), without prior knowledge of the environment.

- As key algorithmic contributions, our proposed method is practically efficient, avoiding both the optimization step of FTRL and the resampling step of FTPL typically required in bandit settings, resulting in significantly faster computation, as shown in our experiments.

- Numerical experiments validate our theoretical results, showing improved regret performance and substantially faster computation compared to the existing BOBW policy.

- We empirically demonstrate that naive combinations of pure exploration policies for exploration and standard exploitation policies degrade performance in the stochastic regime.

## 2 PRELIMINARIES

**Notation** Let $T \in \mathbb{N}$ and $K \in \mathbb{N}$ denote the time horizon and the number of arms, respectively. For $n \in \mathbb{N}$, we use the shorthand $[n] := \{1, \ldots, n\}$. Let $\mathbf{0}$ and $\mathbf{1}$ denote the all-zeros vector and all-ones vector in $\mathbb{R}^K$ and $e_i$ denote the $i$-th standard basis vector in $\mathbb{R}^K$. For an event $A$, we write $\mathbb{1}[A]$ to denote its indicator function, which equals 1 if $A$ occurs and 0 otherwise. We also use the notation $x \wedge y := \min\{x, y\}$.

### 2.1 PROBLEM SETTING

At each round $t \in [T]$, the environment generates a loss vector $\ell_t = (\ell_{t,1}, \ldots, \ell_{t,K}) \in [0,1]^K$, either stochastically or adversarially. In the adversarial regime, loss vectors are determined by either an adaptive adversary, in response to the learner's past actions, or an oblivious adversary, independent of them. In the stochastic regime, by contrast, they are drawn i.i.d. from an unknown but fixed distribution over $[0,1]^K$.

In the decoupled MAB setting, the learner selects an arm $i_t \in [K]$ to exploit, and an arm $j_t \in [K]$ to explore, which may be same or different. Then, the learner suffers $\ell_{t,i_t}$ without observing it, and observes $\ell_{t,j_t}$ without suffering it. The performance of a policy is measured by the pseudo-regret, defined as

$$\mathrm{Reg}(T) = \mathbb{E}\left[\sum_{t=1}^{T} \ell_{t,i_t}\right] - \min_{i \in [K]} \mathbb{E}\left[\sum_{t=1}^{T} \ell_{t,i}\right] = \sum_{t=1}^{T} \mathbb{E}\left[\langle \hat{\ell}_t, w_t - e_{i^*} \rangle\right], \tag{1}$$

where $i^* = \arg\min_{i \in [K]} \mathbb{E}[\sum_{t=1}^{T} \ell_{t,i}]$ denotes the best fixed arm in hindsight, assumed unique. The expectation $\mathbb{E}[\cdot]$ is taken over the randomness of policy and environment. Since only partial feedback is available, the learner constructs an unbiased estimator $\hat{\ell}_t$ of the full loss vector, typically using an importance-weighted (IW) estimator, based on the observed feedback $\ell_{t,j_t}$ of the explored arm. The vector $w_t$ denotes the exploitation probability over arms at round $t$, where $w_{t,i} = \mathbb{P}[i_t = i]$.

We consider the stochastically constrained adversarial (SCA) regime, encompassing the stochastic regime as a special case (Wei & Luo, 2018). In this regime, the environment may adjust the parameters of the arms (e.g., means) over rounds in response to the learner's past actions $\{i_s\}_{s=1}^{t-1}$. However, it is constrained to maintain fixed differences in the expected losses between any pair of arms, i.e., $\mathbb{E}[\ell_{t,i} - \ell_{t,j}] = \Delta_{i,j}$ for all $i, j, t$. Let $\Delta_i = \Delta_{i,i^*}$ denote the suboptimality gap of arm $i$, where $i^* = \arg\min_i \Delta_{i,1}$ holds in this regime. The pseudo-regret in this regime can be expressed as

$$\mathrm{Reg}(T) = \mathbb{E}\left[\sum_{t=1}^{T} \sum_{i \neq i^*} \Delta_i w_{t,i}\right]. \tag{2}$$

### 2.2 PREVIOUS APPROACHES IN DECOUPLED BANDITS

Here, we introduce two representative policies for the decoupled MAB setting. Let $w_{t,i} := \mathbb{P}[i_t = i]$ be the exploitation probability and $p_{t,i} := \mathbb{P}[j_t = i]$ denote the exploration probability of arm $i$ at round $t$. Then, the IW loss estimator is given by $\hat{\ell}_{t,i} = \ell_{t,i} \mathbb{1}[j_t = i] p_{t,i}^{-1}$ and $\hat{L}_{t,i} = \sum_{s=1}^{t-1} \hat{\ell}_{s,i}$ is the estimated cumulative loss up to round $t - 1$.

Avner et al. (2012) proposed a decoupled bandit policy that uses an exploitation strategy based on Exp3 (Auer et al., 2002) and an exploration strategy designed to minimize the variance of the loss

estimates, which scales as $\sum_i w_{t,i}/p_{t,i}$:

$$w_{t,i} = (1-\gamma)\frac{g_{t,i}}{\sum_{j \in [K]} g_{t,j}} + \frac{\gamma}{K} \quad \text{and} \quad p_{t,i} = \frac{\sqrt{w_{t,i}}}{\sum_{j \in [K]} \sqrt{w_{t,i}}}, \tag{3}$$

where $\gamma$ is a parameter that depends on the learning rate $\eta$ and the number of arms $K$. The weight $g_{t,i}$ of each arm is updated according to

$$g_{t+1,i} = \begin{cases} g_{t,i}\exp\left(\eta_{\text{sto}}\hat{\ell}_{t,i} + \frac{\eta\beta}{p_{t,i}}\right), & \text{in the stochastic regime,} \\ g_{t,i}\exp\left(\eta_{\text{adv}}\hat{\ell}_{t,i} + \frac{\eta\beta}{p_{t,i}}\right) + \frac{e}{KT}\sum_{i \in [K]} g_{t,i}, & \text{in the adversarial regime.} \end{cases}$$

Both the above update rule and the learning rate $\eta$ depends on the regime and $T$. This implies that the policy requires prior knowledge of not only the time horizon but also the environment.

For the BOBW guarantee, Rouyer & Seldin (2020) adopted the $\beta$-Tsallis-INF policy with $\beta \in (0,1)$ as the exploitation strategy, a well-known BOBW policy in standard MABs (Zimmert & Seldin, 2021). For exploration, they employed a strategy similar to that of Avner et al. (2012). Together, these form the Decoupled-Tsallis-INF policy:

$$w_t = \underset{w \in \mathcal{S}_{K-1}}{\arg\min}\left\{\left\langle w, \hat{L}_t \right\rangle - \frac{1}{\eta_t}\sum_{i \in [K]}\frac{w_i^\beta - \beta w_i}{\beta(1-\beta)}\right\} \quad \text{and} \quad p_{t,i} = \frac{w_{t,i}^{1-\beta/2}}{\sum_{j \in [K]} w_{t,j}^{1-\beta/2}}. \tag{4}$$

Here, $\mathcal{S}_{K-1} = \{w \in [0,1]^K : \|w\|_1 = 1\}$ denotes the $(K-1)$-dimensional probability simplex and the learning rate is $\eta_t = \mathcal{O}(t^{-1/2})$.[1] This policy achieved BOBW, with $\mathcal{O}(\sqrt{KT})$ adversarial regret for $\beta \in (0,1)$ and time-independent regret $\mathcal{O}(K/\Delta_{\min})$ for $\beta \in (0,2/3)$ in the SCA regime, significantly improving over previous Exp3-type policy. As $\beta \to 1$, $\beta$-Tsallis-INF converges to Exp3[2] and $p_t$ coincides with that of Avner et al. (2012) when $\beta = 1$. In this sense, Decoupled-Tsallis-INF roughly recovers (3) by tuning $\beta$. However, computing $w_t$ in (4) involves a convex optimization step, increasing the overall computational cost as the price for improved regret guarantees.

## 3 PROPOSED POLICY: FTPL FOR DECOUPLED BANDITS

In this section, we elaborate on technical challenges that arise in applying FTPL to decoupled bandits and present our method to overcome them.

### 3.1 TECHNICAL CHALLENGES

A common feature of previous decoupled bandit policies is that the exploration probability $p_{t,i}$ is computed using the exploitation probability $w_{t,i}$. In FTPL, however, $w_{t,i}$ generally lacks a closed-from expression, except in special cases such as FTPL with Gumbel perturbations, where the induced exploitation probability coincides with the multinomial logit model, i.e., the Exp3 policy. Although using Exp3 for exploitation is convenient, it results in suboptimal performance in the stochastic regime (Avner et al., 2012). Instead, to obtain BOBW guarantee with FTPL, it is natural to adopt a Fréchet-type perturbation, due to its correspondence with $\beta$-Tsallis-INF (Kim & Tewari, 2019; Lee et al., 2025), an exploitation strategy known to achieve BOBW, even though $w_{t,i}$ under Fréchet-type perturbations does not have a closed form.

A natural idea is to estimate $w_{t,i}$ via geometric resampling (GR), used in standard MABs to construct the IW estimator (Neu & Bartók, 2016; Chen et al., 2025). However, GR is introduced to estimate only the probability of the selected arm $i_t$, not the full vector $w_t$. This makes direct application infeasible, since computing $p_t$ requires estimates for all arms. Even if one could recover $w_t$ via repeated resampling, the computational cost would increase by a factor of $K$, yielding a per-step cost of at least $\mathcal{O}(K^3)$ or $\mathcal{O}(K^2 \log K)$, depending on the method. Moreover, for arms with very small $w_{t,i}$, the required number of resampling iterations becomes large, as it scales as $1/w_{t,i}$. To overcome these challenges, we propose an alternative method to approximate $w_{t,i}$ without resampling, achieving considerable computational improvement. We describe this method below.

---

[1]Jin et al. (2023) proposed an arm-dependent learning rate that improves the regret order in the SCA regime.
[2]Strictly speaking, it coincides with the version of Exp3 in Bubeck & Cesa-Bianchi (2012), whereas the original one in Auer et al. (2002) (also in Avner et al. (2012)) includes an additional $\gamma/K$ term.

---

**Algorithm 1:** FTPL for decoupled exploration and exploitation

---

**Initialization:** Set $\hat{L}_1 = \mathbf{0}$, shape $\alpha > 1$, and learning rates $\eta_1 \geq \eta_2 \geq \cdots > 0$.

**for** $t = 1$ *to* $T$ **do**
    Sample $(r_{t,1}, \ldots, r_{t,K})$ i.i.d. from $\mathcal{P}_\alpha^K$.         // Pareto perturbation
    Select $i_t = \arg\min_{j \in [K]} \{\hat{L}_{t,j} - r_{t,j}/\eta_t\}$ and incur $\ell_{t,i_t}$.     // exploitation
    Explore $j_t \sim p_t$, where $p_t$ is defined in (7) and observe $\ell_{t,j_t}$.     // exploration
    Update $\hat{L}_{t+1} = \hat{L}_t + \ell_{t,j_t} p_{t,j_t}^{-1} e_{j_t}$.

**end**

---

## 3.2 PROPOSED POLICY

**Exploitation** At each round $t \in [T]$, the learner selects an arm $i_t$ to exploit according to the FTPL policy:

$$i_t = \arg\min_{i \in [K]} \left\{ \hat{L}_{t,i} - \frac{r_{t,i}}{\eta_t} \right\} = \arg\min_{i \in [K]} \left\{ \underline{\hat{L}}_{t,i} - \frac{r_{t,i}}{\eta_t} \right\}, \tag{5}$$

where $\underline{\hat{L}}_t = \hat{L}_t - \mathbf{1} \cdot \min_{i \in [K]} \hat{L}_{t,i} \in [0, \infty)^K$ represents the loss-gap vector and $\eta_t$ denotes the learning rate specified later. Here, $r_t = (r_{t,1}, \ldots, r_{t,K})$ is a random perturbation vector whose components are sampled i.i.d. from the Pareto distribution $\mathcal{P}_\alpha$ with shape parameter $\alpha > 1$. The PDF and CDF of $\mathcal{P}_\alpha$ are

$$f(x) = \frac{\alpha}{x^{\alpha+1}}, \quad F(x) = 1 - \frac{1}{x^\alpha}, \quad x \in [1, \infty),$$

respectively. Then, the exploitation probability of arm $i \in [K]$ given $\underline{\hat{L}}_t$ can be expressed by $w_{t,i} = \phi_i(\eta_t \hat{L}_t)$, where

$$\phi_i(\eta_t \hat{L}_t) := \mathbb{P}_{r_t \sim \mathcal{P}_\alpha^K} \left[ i = \arg\min_{i \in [K]} \left\{ \underline{\hat{L}}_{t,i} - \frac{r_{t,i}}{\eta_t} \right\} \right]$$

$$= \int_1^\infty f(z + \eta_t \underline{\hat{L}}_{t,i}) \prod_{j \neq i} F(z + \eta_t \underline{\hat{L}}_{t,j}) \mathrm{d}z, \tag{6}$$

which cannot be expressed in the closed-form.

**Exploration** In addition to FTPL exploitation, the learner selects an arm $j_t$ for exploration according to the probability distribution $p_t$, defined as

$$p_{t,i} = \frac{q_{t,i}}{\sum_{j \in [K]} q_{t,j}}, \quad \text{where} \quad q_{t,i} = \left( \frac{1}{1 + \eta_t \underline{\hat{L}}_{t,i}} \wedge \frac{1}{\sigma_{t,i}^{1/\alpha}} \right)^{\frac{\alpha+1}{2}}, \tag{7}$$

and $\sigma_{t,i}$ denotes the rank of $\hat{L}_{t,i}$ among $\{\hat{L}_{t,j}\}_{j \in [K]}$, with 1 for the smallest and $K$ for the largest value (ties are broken arbitrarily). It is obvious that $p_t$ is computable directly without additional convex optimization or resampling, at $O(K \log K)$ per-step cost due to sorting $\{\hat{L}_{t,j}\}_j$. Given the correspondence between the $\beta$-Tsallis entropy and Fréchet-type perturbations with $\alpha = 1/(1 - \beta)$, $q_{t,i}$ can be viewed as an approximation of $w_{t,i}^{1/2+1/(2\alpha)}$, which corresponds to $w_{t,i}^{1-\beta/2}$ in (4). Our approach of approximating $w_{t,i}$ using a tight upper bound (see Lemma 10 in Appendix for details) may be of independent interest for efficiently approximating arm-selection probabilities of FTPL beyond the decoupled setting. The pseudo-code of the overall procedure is given in Algorithm 1.

## 3.3 REGRET ANALYSIS

The following results establish the regret guarantees and demonstrate the BOBW property, with the first theorem showing that Algorithm 1 achieves minimax optimality in the adversarial regime.

**Theorem 1.** *In the adversarial regime, Algorithm 1 with $\alpha > 1$ and $\eta_t = cK^{\frac{1}{\alpha} - \frac{1}{2}}/\sqrt{t}$ for $c > 0$ satisfies $\mathrm{Reg}(T) \le \mathcal{O}(\sqrt{KT})$.*

The proof of Theorem 1 is given in Appendix B. This result matches the lower bound of Avner et al. (2012) up to constant factors, and is therefore minimax optimal. In the next theorem, we analyze the regret of Algorithm 1 in the SCA regime, which includes the stochastic regime as a special case.

**Theorem 2.** *In the stochastically constrained adversarial regime with a unique best arm $i^*$, Algorithm 1 with $\alpha \in (1, 3]$ and $\eta_t = cK^{\frac{1}{\alpha} - \frac{1}{2}}/\sqrt{t}$ for $c > 0$ satisfies*

$$\mathrm{Reg}(T) \le \mathcal{O}\left( \left( \sum_{t=1}^{T} \sum_{i \neq i^*} \sqrt{K} \Delta_i^{1-\alpha} t^{-\frac{\alpha}{2}} \right) + \frac{K}{\Delta_{\min}} \right). \tag{8}$$

A proof sketch of Theorem 2 is provided in Section 3.4, with the detailed proof in Appendix C. The theorem focuses on $\alpha \in (1, 3]$, since for $\alpha > 3$ the dependence on $K$ worsens from $\sqrt{K}$ to $K^{\frac{\alpha - 2}{\alpha - 1}}$. Such degradation, which also arises in the BOBW FTRL policy with $\beta \in (2/3, 1)$, is undesirable (Rouyer & Seldin, 2020). Note that the bound in (8) becomes independent of the time horizon $T$ for $\alpha \in (2, 3]$, since $\sum_{t=1}^{T} t^{-\alpha/2}$ converges as $T \to \infty$ whenever $\alpha > 2$. In particular, $\alpha = 3$ minimizes this bound, achieving (near-)optimal regret as follows.

**Corollary 3.** *In the stochastically constrained adversarial regime with a unique best arm $i^*$, Algorithm 1 with $\alpha = 3$ and $\eta_t = cK^{\frac{1}{\alpha} - \frac{1}{2}}/\sqrt{t}$ for $c > 0$ satisfies*

$$\mathrm{Reg}(T) \le \mathcal{O}\left( \sqrt{\frac{K}{\Delta_{\min}} \sum_{i \neq i^*} \frac{1}{\Delta_i}} + \frac{K}{\Delta_{\min}} \right).$$

In general, our bound coincides with Rouyer & Seldin (2020) for $\beta = 2/3$. Given the correspondence $\alpha = 1/(1 - \beta)$ noted earlier, this similarity is natural. However, under specific conditions on the suboptimality gaps, their overall bound can become tighter, as their first term reduces from $\mathcal{O}(K/\Delta_{\min})$ to $\mathcal{O}\left( \sqrt{\frac{K}{\Delta_{\min}} \sum_{i \neq i^*} \frac{1}{\Delta_i}} \right)$. In contrast, our first term already attains their best possible result without any extra assumptions. The relative looseness of our result comes from the analysis of the second term, which scales as $K/\Delta_{\min}$, whereas FTRL achieves the sharper $\sqrt{K}/\Delta_{\min}$ rate. Moreover, we expect that the optimal regret in the SCA regime can be achieved by introducing arm-dependent learning rates, as in FTRL-based methods (Jin et al., 2023), though this would require more intricate analysis techniques beyond the scope of this paper.

### 3.4 PROOF SKETCH OF THE REGRET IN THE SCA REGIME

Here, we provide a proof sketch of Theorem 2. We begin by decomposing the pseudo-regret in (1), which can be seen as a reduction of Lemmas 3.3 and 3.4 in Zhan et al. (2025) from the semi-bandit setting to the MAB setting. For completeness, the detailed proof is given in Appendix A.1.

**Lemma 4** (Regret decomposition). *Let $\{\eta_t\}_{t \in [T]}$ be a sequence of positive, decreasing learning rates and $\eta_0 = \infty$. Then, Algorithm 1 with $\alpha > 1$ satisfies*

$$\mathrm{Reg}(T) \le \sum_{t=1}^{T} \mathbb{E}\left[ \left\langle \hat{\ell}_t, \phi(\eta_t \hat{L}_t) - \phi(\eta_t \hat{L}_{t+1}) \right\rangle \right] + \sum_{t=1}^{T+1} \left( \frac{1}{\eta_t} - \frac{1}{\eta_{t-1}} \right) \mathbb{E}_{r_t \sim \mathcal{P}_\alpha^K}[r_{t,i_t} - r_{t,i^*}]. \tag{9}$$

Following the convention, we refer to the first and second term of (9) as the *stability term* and *penalty term*, respectively. The stability term can be upper bounded as follows.

**Lemma 5.** *For any $t \in [T]$ and $i \in [K]$ Algorithm 1 with $\alpha > 1$ satisfies*

$$\mathbb{E}\left[ \hat{\ell}_{t,i} \left( \phi_i(\eta_t \hat{L}_t) - \phi_i(\eta_t \hat{L}_{t+1}) \right) \Big| \hat{L}_t \right] \le e(\alpha + 1)\eta_t \sum_{j \in [K]} q_{t,j} q_{t,i},$$

*where $q_t$ is defined in (7).*

We provide the proof of Lemma 5 in Appendix A.2. While the proof structure follows Lee et al. (2024), our construction of $p_{t,i}$ allows the stability term to be upper bounded in terms of $q_{t,i}$, which enables the BOBW guarantee in the decoupled setting. The penalty term can be upper bounded as follows, providing a tighter bound than Lee et al. (2024). The proof is provided in Appendix A.3.

**Lemma 6.** *For any $t \in [T]$, Algorithm 1 with $\alpha > 1$ satisfies*

$$\mathbb{E}_{r_t \sim \mathcal{P}_\alpha^K}\left[r_{t,i_t} - r_{t,i^*}\Big|\hat{L}_t\right] \leq \frac{\alpha}{\alpha - 1}\sum_{i \neq i^*}\frac{1}{(1 + \eta_t\underline{\hat{L}}_{t,i})^{\alpha-1}} \wedge C_\alpha K^{\frac{1}{\alpha}},$$

*where $C_\alpha = \frac{2\alpha^3 + (e-2)\alpha^2}{(\alpha-1)(2\alpha-1)}$.*

Under the uniform learning rate in Theorems 1 and 2, the order of the upper bound on the penalty term is never larger than that of the stability term for $\alpha \in (1, 3]$, making the latter dominant in the regret. This observation is consistent with Rouyer & Seldin (2020), who analyze the case $\beta \in (0, 2/3]$. As discussed in Corollary 3, one may design arm-dependent (Jin et al., 2023) or stability–penalty matching (Tsuchiya et al., 2023; Ito et al., 2024) learning rates, instead of a uniform learning rate, to equalize contributions of the two terms. This could yield (possibly improved) BOBW guarantees for general $\alpha > 1$. However, to the best of our knowledge, such designs have not been explored for FTPL differently from FTRL frameworks, primarily because $w_{t,i}$ lacks a closed-form expression.

**Proof sketch**  In the SCA regime, the regret is expressed in terms of the suboptimality gap $\Delta_i$ and exploitation probability $w_{t,i}$. Therefore, to derive a meaningful bound, we express the stability and penalty terms in terms of $w_{t,i}$ and $\Delta_i$. Adopting the approach of previous FTPL analyses in standard MABs (Honda et al., 2023; Lee et al., 2024), we define the following events:

$$D_t := \left\{\sum_{i \neq i^*}\frac{1}{(2^{1/\alpha} + \eta_t\underline{\hat{L}}_{t,i})^\alpha} \leq \frac{1}{2}\right\}. \tag{10}$$

When $D_t$ occurs, it implies that $\underline{\hat{L}}_{t,i^*} = 0$, indicating that the optimal arm $i^*$ has been accurately identified based on the information so far. On $D_t$, we can also upper bound $q_{t,i}$ in terms of $w_{t,i}$,

$$q_{t,i} \leq \left(\frac{1}{1 + \eta_t\underline{\hat{L}}_{t,i}}\right)^{\frac{\alpha+1}{2}} \leq (2e^2 w_{t,i})^{\frac{1}{2} + \frac{1}{2\alpha}} \leq 2e^2(w_{t,i})^{1 - \frac{1}{\alpha}}, \forall i \neq i^*,$$

where the second step follows from Lemma 10 and the last step holds when $\alpha \in (1, 3]$. Note that the design of $D_t$ enables us to establish an explicit relationship between $q_{t,i}$ and $w_{t,i}$, which may be of independent interest beyond the decoupled setting. In Appendix D, we show that the contribution of the optimal arm $i^*$ to the stability term is bounded by the total contribution of the suboptimal arms on $D_t$, i.e., $\mathcal{O}(\eta_t\sum_{j\in[K]}q_{t,j}\cdot\sum_{i\neq i^*}q_{t,i})$. Therefore, by Lemma 5, we have

$$\text{Reg}(T) \leq \mathbb{E}\left[\sum_{t=1}^T\mathcal{O}\left(\mathbb{1}[D_t]\frac{K^{\frac{1}{\alpha} - \frac{1}{2}}}{\sqrt{t}}\sum_{j\in[K]}q_{t,j}\sum_{i\neq i^*}w_{t,i}^{1 - \frac{1}{\alpha}}\right) + \mathcal{O}\left(\mathbb{1}[D_t^c]\sqrt{K/t}\right)\right]$$

$$\leq \mathbb{E}\left[\sum_{t=1}^T\mathcal{O}\left(\mathbb{1}[D_t]\frac{K^{\frac{1}{2\alpha}}}{\sqrt{t}}\sum_{i\neq i^*}w_{t,i}^{1 - \frac{1}{\alpha}}\right) + \mathcal{O}\left(\mathbb{1}[D_t^c]\sqrt{K/t}\right)\right],$$

where the last step follows from the definition of $q_{t,i}$, which implies $\sum_i q_{t,i} \leq \sum_i i^{-\frac{1}{2} - \frac{1}{2\alpha}} \leq \frac{2\alpha}{\alpha - 1}K^{\frac{1}{2} - \frac{1}{2\alpha}}$. By the definition of pseudo-regret in the SCA regime in (2), we obtain

$$\text{Reg}(T) \geq \mathbb{E}\left[\sum_{t=1}^T\mathbb{1}[D_t]\sum_{i\neq i^*}\Delta_i w_{t,i} + \Omega(\mathbb{1}[D_t^c]\Delta_{\min})\right],$$

where $\Omega$ denotes the big-Omega notation. By applying the self-bounding technique, we have

$$\text{Reg}(T) \leq \mathbb{E}\left[\sum_{t=1}^T\mathcal{O}\left(\mathbb{1}[D_t]\left(\sum_{i\neq i^*}\frac{K^{\frac{1}{2\alpha}}}{\sqrt{t}}w_{t,i}^{1 - \frac{1}{\alpha}} - \Delta_i w_{t,i}\right) + \mathbb{1}[D_t^c]\left(\sqrt{K/t} - \Delta_{\min}\right)\right)\right].$$

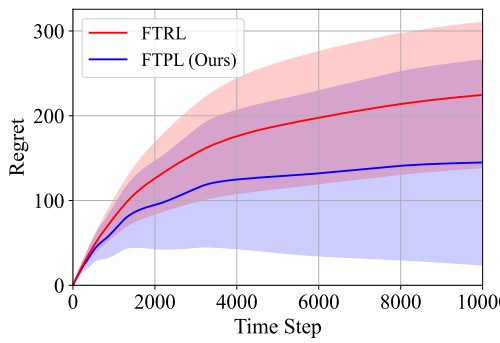 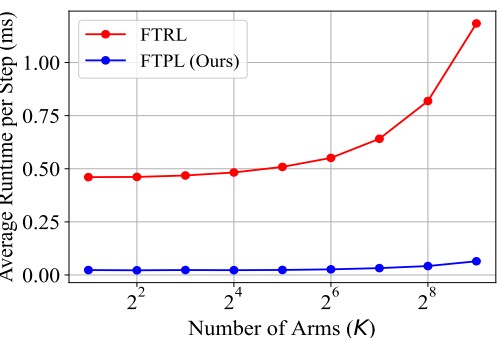

Figure 1: Adversarial regret with $\Delta = 0.125$     Figure 2: Computation time (ms)

Since $\max_{w \in [0,1]} \frac{A w^{1-\frac{1}{\alpha}}}{\sqrt{t}} - \Delta_i w \leq \mathcal{O}(A^\alpha \Delta_i^{1-\alpha} t^{-\alpha/2})$, the regret satisfies

$$\mathrm{Reg}(T) \leq \mathcal{O}\left( \left( \sum_{t=1}^{T} \sum_{i \neq i^*} \sqrt{K} \Delta_i^{1-\alpha} t^{-\frac{\alpha}{2}} \right) + \frac{K}{\Delta_{\min}} \right).$$

## 4   NUMERICAL EXPERIMENTS

We evaluate the empirical performance and computational efficiency of our policy, Algorithm 1, under two regimes in the decoupled setting: adversarial and stochastic regimes. All experiments are conducted for 1000 independent repetitions unless otherwise specified, with the time horizon $T = 10000$, $\alpha = 3$ for Algorithm 1, and $\beta = 2/3$ for Decoupled-Tsallis-INF (Rouyer & Seldin, 2020). For simplicity, we denote Algorithm 1 by FTPL and Decoupled-Tsallis-INF by FTRL in this section. Although the constant $c$ in the learning rate $\eta_t$ can be tuned either to minimize regret bound analytically or empirically, we set $c = 2$ following prior studies (Zimmert & Seldin, 2021; Honda et al., 2023; Lee et al., 2024). The shaded regions in the figures indicate one standard deviation.

**Adversarial regime**  For the adversarial regime, we follow the setup of Zimmert & Seldin (2021). Specifically, the mean loss of the optimal arm and all suboptimal arms alternates between $(0, \Delta)$ and $(1 - \Delta, 1)$, with the duration of each phase growing exponentially as $\lfloor 1.6^n \rfloor$, where $n$ denotes the phase index. We consider an eight-armed bandit with a unique optimal arm under $\Delta = 0.125$ and $\Delta = 0.0625$, with results for the latter provided in Appendix E.

The first experiment (Figures 1 and 2) evaluates FTRL and FTPL with $\Delta = 0.125$. Specifically, Figure 1 presents the empirical performance of FTPL in comparison to FTRL, showing that our policy achieves lower cumulative regret with sublinear growth.

Figure 2 shows the computational efficiency of FTPL relative to FTRL. The average per-step runtime is measured over 100 independent repetitions as the number of arm increases, with $K \in \{2^i : i \in [9]\}$, where the $x$-axis is displayed in logarithmic scale. While the FTRL policy requires solving a convex optimization problem, in the Tsallis entropy case, $p_t$ admits a closed-form expression, which we efficiently compute using Newton's method (Zimmert & Seldin, 2021). To further demonstrate the efficiency of Newton's method, we evaluate the per-step runtime of FTRL using a splitting conic solver for $K \in \{2^i : i \in [6]\}$ in Appendix E (O'Donoghue et al., 2016). Even with this efficient computation, Figure 2 shows that the runtime of FTRL grows rapidly with $K$, whereas FTPL remains nearly constant. Over the evaluated range of arms, the per-step runtime of FTRL is roughly 20 times greater than that of FTPL, showing the substantial computational advantage of our policy.

**Stochastic regime**  For the stochastic regime, we adopt the setup of Jourdan et al. (2023), considering the five-armed bandit with a unique optimal arm, where each arm provides Bernoulli rewards with mean loss vector $\mu = (0.4, 0.45, 0.55, 0.7, 0.8)$.

The second experiment (Figure 3) compares the empirical performance of EB-TC (Jourdan et al., 2023), FTRL, and FTPL. EB-TC is an anytime sampling rule for pure exploration tasks that does

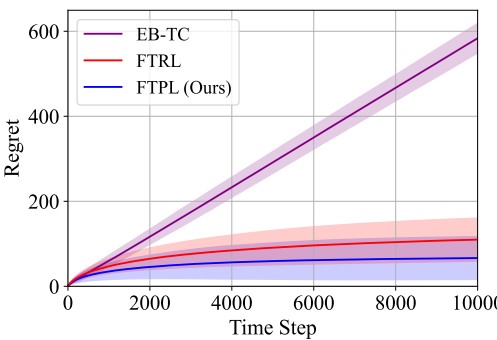 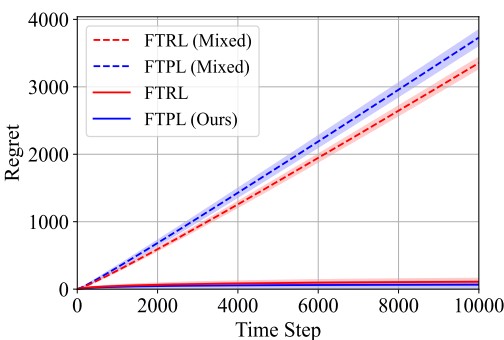

Figure 3: Stochastic regret with EB-TC, a pure exploration policy attaining constant cumulative regret. See Section 4 for details.

Figure 4: Stochastic regret of mixed policies, which select the explored arm via EB-TC and exploited arm according to FTPL or FTRL.

not require a predefined time horizon or confidence level. While it aims to minimize the expected simple regret in the standard MAB, it has been shown to achieve constant cumulative regret in the decoupled MAB, allowing a direct comparison with our policy. As shown in Figure 3, our policy achieves nearly-constant cumulative regret, outperforming both FTRL, as it did in the adversarial regime, and EB-TC. The suboptimal performance of EB-TC is consistent with its regret order of $\tilde{\mathcal{O}}(K^3/\Delta_{\min}^2)$, compared to $\mathcal{O}(K/\Delta_{\min})$ for FTRL and FTPL.

The third experiment (Figure 4) evaluates the performance of the mixed policies FTPL(Mixed) and FTRL(Mixed), where EB-TC is used for exploration (i.e., $j_t \sim$ EB-TC), while $i_t$ is sampled according to (5) and (4), respectively. Rouyer & Seldin (2020) conjectured that directly combining a pure exploration policy with a standard bandit policy for exploitation is suboptimal, which our experiment confirms: the cumulative regret of the mixed policies grows substantially faster than FTPL and FTRL alone, empirically demonstrating their suboptimality. This result aligns with our analysis, which relies on controlling $-\phi_i'/p_{t,i}$, where $-\phi_i'(\lambda) = \partial\phi_i(\lambda)/\partial\lambda_i$, a quantity central to BOBW guarantees (Abernethy et al., 2015; Bubeck, 2019; Lee et al., 2025). Since $\phi_i'$ is the derivative of $w_{t,i}$, the link between $w_{t,i}$ and $p_{t,i}$ is crucial, as in standard MABs where $p_{t,i} = w_{t,i}$. In our design, we define $p_t$ as the normalization of approximations to $w_{t,i}^{1/2+1/(2\alpha)}$, thereby preserving this coupling and ensuring desired bounds. In contrast, replacing $p_t$ with a pure exploration policy (usually deterministic) breaks this coupling, possibly making $p_t$ one-hot vector and inflating $-\phi_i'/p_{t,i}$ for $i \neq j_t$, which invalidates the analysis and can increase the cumulative regret.

## 5 CONCLUSION

We proposed a practically efficient FTPL policy with Pareto perturbations that guarantees BOBW in the decoupled MAB problem. Our policy achieves minimax optimal regret $\mathcal{O}(\sqrt{KT})$ in the adversarial regime, and a near-optimal time-independent regret bound of $\mathcal{O}(K/\Delta_{\min})$ in the stochastically constrained adversarial regime, where $\Delta_{\min} = \min_{i \neq i^*} \Delta_i$. Our result improves upon the optimal regret bound of the standard stochastic MAB, $\mathcal{O}(\sum_{i \neq i^*} \log T/\Delta_i)$, which is time-dependent.

In addition to these theoretical strengths, we avoid both the convex optimization step in FTRL, and the resampling step typically required in FTPL for estimating arm-selection probabilities. As a result, our policy runs about 20 times faster than Decoupled-Tsallis-INF, while achieving better empirical performance across both regimes. Furthermore, we empirically showed that our policy outperforms the pure exploration policy, which is known to yield time-independent cumulative regret in the decoupled stochastic MAB. These findings offer insight into the design of refined learning rates (e.g., adaptive learning rates) for FTPL: by using an approximation of $w_t$, one can adjust the learning rate in a way analogous to FTRL frameworks, where the learning rate is explicitly determined by $w_t$. This perspective potentially serves as a foundation for establishing BOBW guarantees for FTPL beyond the MAB setting. Finally, we empirically confirmed that naively mixing a pure exploration policy and a standard bandit policy for exploitation is suboptimal, implying the necessity of dedicated policies for decoupled settings.

## LLM USAGE

Large Language Models (LLMs) were only used for polishing and minor grammar corrections. No part of the analysis, derivations, or core writing was generated by LLMs.

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

# A    OMITTED PROOFS FOR LEMMAS

In this section, we provide the detailed proofs for lemmas omitted in the main paper.

## A.1    PROOF FOR THE REGRET DECOMPOSITION (LEMMA 4)

While the overall proof is a straightforward adaptation of the arguments in the semi-bandit setting, particularly Lemmas 3.3 and 3.4 from Zhan et al. (2025), to the MAB setting, we provide all details here for completeness.

We begin by recalling the connection between FTPL and FTRL, where it is known that FTPL can generally be expressed as FTRL with a specific corresponding regularizer (Abernethy et al., 2015; Suggala & Netrapalli, 2020). To formalize this, consider a convex potential function $\Phi : \mathbb{R}^K \to \mathbb{R}$ for $\phi$ defined as

$$\Phi(\lambda) = \mathbb{E}_r\left[\max_{i\in[K]}\{\lambda_i + r_i\}\right],$$

so that the gradient of $\Phi$ satisfies $\nabla\Phi(\lambda) = \phi(-\lambda)$, where $\phi$ denotes the arm-selection probability function of FTPL. The convex conjugate (or Lagrange transform) of $\Phi$ is given by

$$\Phi^*(p) = \sup_{\lambda\in\mathbb{R}^K} \langle p, \lambda\rangle - \Phi(\lambda), \quad \text{for } p \in \text{Int}(\mathcal{P}_{K-1}),$$

where $\text{Int}(\mathcal{P}_{K-1})$ denotes the interior of the probability simplex of dimension $K-1$. It is known that FTPL is equivalent to FTRL with regularizer $\Phi^*(p)$. By standard results in the convex analysis (see Zhan et al., 2025, Lemma G.1 and the references therein), one can see that $w_t = \nabla\Phi(-\eta_t \hat{L}_t)$ implies $-\eta_t \hat{L}_t \in \partial\Phi^*(w_t)$, and hence

$$w_t \in \underset{x\in\mathcal{P}_{K-1}}{\arg\min} \left\{\Phi^*(x)/\eta_t + \left\langle x, \hat{L}_t\right\rangle\right\}.$$

It is worth noting that if $w_t$ lies on the boundary of the simplex, the gradient $\nabla\Phi^*(p)$ may not exist. Nevertheless, we consider minimization over $\mathcal{P}_{K-1}$ rather than $\text{Int}(\mathcal{P}_{K-1})$, since the regularizer $\Phi^*(p)$ remains well-defined even on boundary points, although its gradients may be unbounded.

**Lemma 4 (restated)** *Let $\{\eta_t\}_{t\in[T]}$ be a sequence of positive, decreasing learning rates and $\eta_0 = \infty$. Then, Algorithm 1 with $\alpha > 1$ satisfies*

$$\text{Reg}(T) \leq \sum_{t=1}^{T}\mathbb{E}\left[\left\langle \hat{\ell}_t, \phi(\eta_t\hat{L}_t) - \phi(\eta_t\hat{L}_{t+1})\right\rangle\right] + \sum_{t=1}^{T+1}\left(\frac{1}{\eta_t} - \frac{1}{\eta_{t-1}}\right)\mathbb{E}_{r_t\sim\mathcal{D}}[r_{t,i_t} - r_{t,i^*}].$$

*Proof.* Following the proof of Lemma 3.3 of Zhan et al. (2025), let $\Phi_t^*(x) = \Phi^*(x)/\eta_t + \left\langle x, \hat{L}_t\right\rangle$. By definition, we have $w_t \in \arg\min_{x\in\mathcal{P}_{K-1}} \Phi_t^*(x)$ and

$$\sum_{t=1}^{T}\left\langle w_t - e_{i^*}, \hat{\ell}_t\right\rangle$$

$$= \sum_{t=1}^{T}\left\langle w_t - w_{t+1}, \hat{\ell}_t\right\rangle + \sum_{t=1}^{T}\left\langle w_{t+1}, \hat{\ell}_t\right\rangle - \sum_{t=1}^{T}\left\langle e_{i^*}, \hat{\ell}_t\right\rangle$$

$$= \sum_{t=1}^{T}\left\langle w_t - w_{t+1}, \hat{\ell}_t\right\rangle + \sum_{t=1}^{T}\left(\Phi_{t+1}^*(w_{t+1}) - \frac{\Phi^*(w_{t+1})}{\eta_{t+1}} - \left(\Phi_t^*(w_{t+1}) - \frac{\Phi^*(w_{t+1})}{\eta_t}\right)\right)$$

$$- \sum_{t=1}^{T}\left(\Phi_{t+1}^*(e_{i^*}) - \frac{\Phi^*(e_{i^*})}{\eta_{t+1}} - \left(\Phi_t^*(e_{i^*}) - \frac{\Phi^*(e_{i^*})}{\eta_t}\right)\right) \tag{11}$$

$$= \sum_{t=1}^{T}\left\langle w_t - w_{t+1}, \hat{\ell}_t\right\rangle + \sum_{t=1}^{T}\left(\Phi_t^*(w_t) - \Phi_t^*(w_{t+1})\right) + \sum_{t=2}^{T+1}\left(\frac{1}{\eta_t} - \frac{1}{\eta_{t-1}}\right)\left(\Phi^*(e_{i^*}) - \Phi^*(w_t)\right)$$

$$+ \Phi_{T+1}^*(w_{T+1}) - \Phi_1^*(w_1) - \Phi_{T+1}^*(e_{i^*}) + \Phi_1^*(e_{i^*})$$

where (11) follows from the definition of $\hat{L}_t$ and $\Phi_t^*$ that

$$\left\langle w_{t+1}, \hat{\ell}_t \right\rangle = \left\langle w_{t+1}, \hat{L}_{t+1} - \hat{L}_t \right\rangle \quad \text{and} \quad \left\langle x, \hat{L}_t \right\rangle = \Phi_t^*(x) - \Phi^*(x)/\eta_t.$$

Since $\hat{L}_1 = \mathbf{0}$ and $\Phi_{T+1}^*(w_{T+1}) \le \Phi_{T+1}^*(e_{i^*})$, we have

$$\sum_{t=1}^{T} \left\langle w_t - e_{i^*}, \hat{\ell}_t \right\rangle \le \sum_{t=1}^{T} \left( \left\langle w_t - w_{t+1}, \hat{\ell}_t \right\rangle + \Phi_t^*(w_t) - \Phi_t^*(w_{t+1}) \right)$$

$$+ \sum_{t=2}^{T+1} \left( \frac{1}{\eta_t} - \frac{1}{\eta_{t-1}} \right) (\Phi^*(e_{i^*}) - \Phi^*(w_t)) + \frac{\Phi^*(e_{i^*}) - \Phi^*(w_1)}{\eta_1}.$$

For notational simplicity, let $\eta_0 = \infty$, which is not used in the policy and is introduced merely for the analysis. Then,

$$\sum_{t=1}^{T} \left\langle w_t - e_{i^*}, \hat{\ell}_t \right\rangle \le \sum_{t=1}^{T} \left( \left\langle w_t - w_{t+1}, \hat{\ell}_t \right\rangle + \Phi_t^*(w_t) - \Phi_t^*(w_{t+1}) \right)$$

$$+ \sum_{t=1}^{T+1} \left( \frac{1}{\eta_t} - \frac{1}{\eta_{t-1}} \right) (\Phi^*(e_{i^*}) - \Phi^*(w_t)). \tag{12}$$

Let $D_\Phi$ denote the Bregman divergence associated with $\Phi$, which is defined by

$$D_\Phi(x, y) = \Phi(x) - \Phi(y) - \langle \nabla\Phi(y), x - y \rangle, \ \forall x, y \in \mathbb{R}^K.$$

Since $w_t \in \partial\Phi^*(-\eta_t \hat{L}_t)$ and $-\eta_t \hat{L}_t \in \partial\Phi(w_t)$, it holds

$$\Phi(-\eta_t \hat{L}_t) + \Phi^*(w_t) = \left\langle w_t, -\eta_t \hat{L}_t \right\rangle.$$

for all $t \in \mathbb{N}$. Then, we have

$$\Phi_t^*(w_t) - \Phi_t^*(w_{t+1}) = -\frac{1}{\eta_t} \left( \Phi^*(w_{t+1}) - \Phi^*(w_t) - \left\langle w_{t+1} - w_t, -\eta_t \hat{L}_t \right\rangle \right)$$

$$\text{(by definition of } \Phi_t^*)$$

$$= -\frac{1}{\eta_t} \left( \left\langle w_{t+1}, -\eta_{t+1} \hat{L}_{t+1} \right\rangle - \Phi(-\eta_{t+1} \hat{L}_{t+1}) \right.$$

$$\left. + \left\langle w_t, \eta_t \hat{L}_t \right\rangle + \Phi(-\eta_t \hat{L}_t) - \left\langle w_{t+1} - w_t, -\eta_t \hat{L}_t \right\rangle \right)$$

$$= -\frac{1}{\eta_t} \left( \Phi(-\eta_t \hat{L}_t) - \Phi(-\eta_{t+1} \hat{L}_{t+1}) - \left\langle w_{t+1}, -\eta_t \hat{L}_t + \eta_{t+1} \hat{L}_{t+1} \right\rangle \right)$$

$$= -\frac{1}{\eta_t} D_\Phi(-\eta_t \hat{L}_t, -\eta_{t+1} \hat{L}_{t+1}). \qquad (\because w_{t+1} = \nabla\Phi(-\eta_{t+1} \hat{L}_{t+1}))$$

On the other hand, by definition, one can obtain (or see Lemma G.2 of Zhan et al. (2025))

$$D_\Phi(x, y) + D_\Phi(z, x) - D_\Phi(z, y) = \langle \nabla\Phi(x) - \nabla\Phi(y), x - z \rangle$$

for any $x, y, z \in \mathbb{R}^K$. Therefore, by letting $x = -\eta_t \hat{L}_{t+1}$, $y = -\eta_{t+1} \hat{L}_{t+1}$ and $z = -\eta_t \hat{L}_t$, we obtain

$$D_\Phi(-\eta_t \hat{L}_{t+1}, -\eta_{t+1} \hat{L}_{t+1}) + D_\Phi(-\eta_t \hat{L}_t, -\eta_t \hat{L}_{t+1}) - D_\Phi(-\eta_t \hat{L}_t, -\eta_{t+1} \hat{L}_{t+1})$$

$$= \left\langle \nabla\Phi(-\eta_t \hat{L}_{t+1}) - w_{t+1}, -\eta_t \hat{L}_{t+1} + \eta_t \hat{L}_t \right\rangle$$

$$= \left\langle \phi(-\eta_t \hat{L}_{t+1}) - w_{t+1}, -\eta_t \hat{\ell}_t \right\rangle,$$

which implies

$$\left\langle w_t - w_{t+1}, \hat{\ell}_t \right\rangle + \Phi_t^*(w_t) - \Phi_t^*(w_{t+1})$$

$$= \frac{1}{\eta_t} \left\langle w_t - w_{t+1}, \eta_t \hat{\ell}_t \right\rangle - \frac{1}{\eta_t} D_\Phi(-\eta_t \hat{L}_t, -\eta_{t+1} \hat{L}_{t+1})$$

$$= \frac{1}{\eta_t} \left\langle w_t - \phi(-\eta_t \hat{L}_{t+1}) + \phi(-\eta_t \hat{L}_{t+1}) - w_{t+1}, \eta_t \hat{\ell}_t \right\rangle - \frac{1}{\eta_t} D_\Phi(-\eta_t \hat{L}_t, -\eta_{t+1} \hat{L}_{t+1})$$

$$= \left\langle w_t - \phi(-\eta_t \hat{L}_{t+1}), \hat{\ell}_t \right\rangle + \frac{1}{\eta_t} \left( \left\langle \phi(-\eta_t \hat{L}_{t+1}) - w_{t+1}, \eta_t \hat{\ell}_t \right\rangle - D_\Phi(-\eta_t \hat{L}_t, -\eta_{t+1} \hat{L}_{t+1}) \right)$$

$$= \left\langle w_t - \phi(-\eta_t \hat{L}_{t+1}), \hat{\ell}_t \right\rangle - \frac{1}{\eta_t} \left( D_\Phi(-\eta_t \hat{L}_{t+1}, -\eta_{t+1} \hat{L}_{t+1}) + D_\Phi(-\eta_t \hat{L}_t, -\eta_t \hat{L}_{t+1}) \right)$$

$$\leq \left\langle w_t - \phi(\eta_t \hat{L}_{t+1}), \hat{\ell}_t \right\rangle. \qquad\qquad (\because D_\Phi(\cdot, \cdot) \geq 0)$$

Since $w_t = \phi(\eta_t \hat{L}_t)$, it remains to control the second term in (12).

While it can be obtained by direct application of Lemma 3.4 of Zhan et al. (2025), we provide the corresponding proof here for completeness. By definition of $\Phi^*$ and $w_t = \nabla \Phi(-\eta_t \hat{L}_t)$, we have

$$\Phi^*(w_t) = -\left\langle \eta_t \hat{L}_t, w_t \right\rangle - \Phi(-\eta_t \hat{L}_t) = -\mathbb{E}\left[ \left\langle \eta_t \hat{L}_t, e_{i_t} \right\rangle \right] + \mathbb{E}\left[ \min_i \eta_t \hat{L}_{t,i} - r_{t,i} \right]$$

$$= -\mathbb{E}\left[ \left\langle \eta_t \hat{L}_t, e_{i_t} \right\rangle \right] + \mathbb{E}\left[ \left\langle \eta_t \hat{L}_t - r_t, e_{i_t} \right\rangle \right]$$

$$= -\mathbb{E}[r_{t,i_t}].$$

By definition of $\Phi$, we have $\Phi(\lambda) \geq \mathbb{E}_r[\langle r + \lambda, p \rangle]$ for any $p \in \mathcal{P}_{K-1}$ and $\lambda \in \mathbb{R}^K$. Hence,

$$\Phi^*(e_{i^*}) = \sup_{x \in \mathbb{R}^K} \langle x, e_i^* \rangle - \Phi(x) \leq \sup_{x \in \mathbb{R}^K} \langle x, e_i^* \rangle - \mathbb{E}_r[\langle r + x, e_{i^*} \rangle]$$

$$= -\mathbb{E}_r[\langle r, e_{i^*} \rangle] = -\mathbb{E}_r[r_{i^*}],$$

which concludes the proof. $\qquad\qquad\qquad\qquad\qquad\qquad\qquad\qquad\qquad\qquad\qquad\qquad\qquad\square$

### A.2 PROOF FOR THE STABILITY TERM (LEMMA 5)

**Lemma 5 (restated)** *For any $t \in [T]$, $i \in [K]$, Algorithm 1 with $\alpha > 1$ satisfies*

$$\mathbb{E}\left[ \hat{\ell}_{t,i} \left( \phi_i(\eta_t \hat{L}_t) - \phi_i(\eta_t \hat{L}_{t+1}) \right) \Big| \hat{L}_t \right] \leq e(\alpha + 1)\eta_t \sum_{j \in [K]} q_{t,j} q_{t,i},$$

*where $q_t$ is defined in (7).*

*Proof.* Let $\lambda \in \mathbb{R}^K$ and $\phi_i'(\lambda) = \frac{\partial \phi_i(\lambda)}{\partial \lambda_i}$, which is

$$\phi_i'(\lambda) = \int_{-\min_j \lambda_j}^{\infty} -\frac{\alpha(\alpha+1)}{(z + \lambda_i + 1)^{\alpha+2}} \prod_{j \neq i} \left( 1 - \frac{1}{(z + \lambda_j + 1)^\alpha} \right) \mathrm{d}z$$

$$= \int_0^{\infty} -\frac{\alpha(\alpha+1)}{(z + \underline{\lambda}_i + 1)^{\alpha+2}} \prod_{j \neq i} \left( 1 - \frac{1}{(z + \underline{\lambda}_j + 1)^\alpha} \right) \mathrm{d}z,$$

where the underline denotes $\underline{\lambda} = \lambda - \mathbf{1} \cdot \min_j \lambda_j$. Note that $-\phi_i'(\lambda)$ is decreasing with respect to $\lambda_i$ and increasing with respect to $\lambda_j$. Then, by definition, we have

$$\mathbb{1}[i = j_t] \left( \phi_i(\eta_t \hat{L}_t) - \phi_i(\eta_t \hat{L}_{t+1}) \right)$$

$$= \mathbb{1}[i = j_t] \int_0^{\eta_t \ell_{t,i} p_{t,i}^{-1}} -\phi_i'(\eta_t \hat{L}_t + x e_i) \mathrm{d}x$$

$$= \mathbb{1}[i = j_t] \int_0^{\eta_t \ell_{t,i} p_{t,i}^{-1}} -\phi_i'(\eta_t \hat{L}_t) \mathrm{d}x \qquad\qquad (\because \text{decreasing w.r.t. } \lambda_i)$$

$$= \mathbb{1}[i = j_t] \int_0^{\infty} \frac{\alpha(\alpha+1)\eta_t \ell_{t,i} p_{t,i}^{-1}}{(z + \underline{\lambda}_i + 1)^{\alpha+2}} \prod_{j \neq i} \left( 1 - \frac{1}{(z + \underline{\lambda}_j + 1)^\alpha} \right) \mathrm{d}z.$$

Let $I_{i,\alpha+2}(\lambda) = \int_0^\infty \frac{1}{(z+\lambda_i+1)^{\alpha+2}} \prod_{j\neq i} \left(1 - \frac{1}{(z+\lambda_j+1)^\alpha}\right) dz$. Then,

$$
\mathbb{E}\left[\hat{\ell}_{t,i}\left(\phi_i(\eta_t \hat{L}_t) - \phi_i(\eta_t \hat{L}_{t+1})\right)\Big|\hat{L}_t\right] = \mathbb{E}\left[\frac{\ell_{t,i}\mathbb{1}[j_t=i]}{p_{t,i}}\left(\phi_i(\eta_t \hat{L}_t) - \phi_i(\eta_t \hat{L}_{t+1})\right)\Big|\hat{L}_t\right]
$$

$$
\leq \alpha(\alpha+1)\eta_t \mathbb{E}\left[\frac{\ell_{t,i}^2 \mathbb{1}[j_t=i]}{p_{t,i}^2} I_{i,\alpha+2}(\eta_t \underline{\hat{L}}_t)\Big|\hat{L}_t\right]
$$

$$
\leq \alpha(\alpha+1)\eta_t \mathbb{E}\left[\frac{I_{i,\alpha+2}(\eta_t \underline{\hat{L}}_t)}{p_{t,i}}\Big|\hat{L}_t\right],
$$

where the last inequality follows from $\ell_{t,i} \leq 1$ and $\mathbb{E}[\mathbb{1}[j_t=i]|\hat{L}_t] = p_{t,i}$.

By definition of $I_{i,\alpha+2}$, one can see that

$$
I_{i,\alpha+2}(\underline{\lambda}) \leq I_{i,\alpha+2}(\lambda^*), \quad \text{where} \quad \lambda_j^* = \begin{cases} \underline{\lambda}_i, & \sigma_j \leq \sigma_i, \\ \infty, & \sigma_j > \sigma_i, \end{cases}
$$

where $\sigma_i$ denotes the rank of $\lambda_i$ in the increasing order of $\lambda$, i.e., $\sigma_i < \sigma_j$ iff $\lambda_i \leq \lambda_j$ with arbitrary tie-breaking rule. In the later of the proof, we assume $\sigma_i = i$ without loss of generality for the simplicity, i.e., $\lambda_1 \leq \lambda_2 \leq \ldots, \lambda_K$. Then, we have

$$
I_{i,\alpha+2}(\lambda^*) = \int_0^\infty \frac{1}{(z + \underline{\lambda}_i + 1)^{\alpha+2}} \left(1 - \frac{1}{(z + \underline{\lambda}_i + 1)^\alpha}\right)^{i-1} dz
$$

$$
= \frac{1}{\alpha} \int_0^{\frac{1}{(1+\underline{\lambda}_i)^\alpha}} w^{\frac{1}{\alpha}}(1-w)^{i-1} dw
$$

$$
= \frac{1}{\alpha} B\left(\frac{1}{(1+\underline{\lambda}_i)^\alpha}; 1 + \frac{1}{\alpha}, i\right),
$$

where $B(x; a, b) = \int_0^x t^{a-1}(1-t)^{b-1} dt$ denotes the incomplete Beta function. By elementary calculation, we obtain for any $x \in [0,1]$

$$
B\left(x; 1 + \frac{1}{\alpha}, i\right) = \int_0^x t^{\frac{1}{\alpha}}(1-t)^{i-1} dt \leq \int_0^x t^{\frac{1}{\alpha}} e^{-t(i-1)} dt
$$

$$
\leq e \int_0^x t^{\frac{1}{\alpha}} e^{-ti} dt \qquad (\because x \in [0,1])
$$

$$
= \frac{e}{i^{1+\frac{1}{\alpha}}} \gamma\left(1 + \frac{1}{\alpha}, xi\right),
$$

where $\gamma(a, x)$ denotes the lower incomplete gamma function. Since $\gamma(a, x) \leq \Gamma(a)$ for any $x > 0$, we have

$$
I_{i,\alpha+2}(\lambda^*) \leq \frac{e}{\alpha i^{1+\frac{1}{\alpha}}} \Gamma\left(1 + \frac{1}{\alpha}\right) \leq \frac{e}{\alpha i^{1+\frac{1}{\alpha}}} \Gamma(2) = \frac{e}{\alpha i^{1+\frac{1}{\alpha}}}. \tag{13}
$$

On the other hand, by Equation 8.10.2 of Olver (2010), it holds that

$$
\gamma\left(1 + \frac{1}{\alpha}, \frac{i}{(1+\underline{\lambda}_i)^\alpha}\right) \leq \frac{\alpha}{\alpha+1} \frac{i^{1/\alpha}}{(1+\underline{\lambda}_i)} \min\left(1, \frac{i}{(1+\underline{\lambda}_i)^\alpha}\right),
$$

which implies

$$
I_{i,\alpha+2}(\lambda^*) \leq \frac{e}{\alpha+1} \frac{1}{(1+\underline{\lambda}_i)i}. \tag{14}
$$

Therefore, from (13) and (14), we obtain

$$
\alpha(\alpha+1)\mathbb{E}\left[\frac{I_{i,\alpha+2}(\eta_t \underline{\hat{L}}_t)}{p_{t,i}}\Big|\hat{L}_t\right] \leq \frac{e(\alpha+1)}{p_{t,i}i^{1+\frac{1}{\alpha}}} \wedge \frac{e\alpha}{p_{t,i}(1+\eta_t \hat{L}_{t,i})i}.
$$

By definition of $p_t$ and $q_{t,i}$ in (7), when $\frac{1}{(1+\eta_t \hat{\underline{L}}_{t,i})} \leq \frac{1}{i^{1/\alpha}}$, we have

$$\frac{1}{p_{t,i}} \frac{1}{(1+\eta_t \hat{\underline{L}}_{t,i})i} = \sum_j q_{t,j} \frac{\sqrt{i(1+\eta_t \hat{\underline{L}}_{t,i})}}{(1+\eta_t \hat{\underline{L}}_{t,i})i} = \sum_j q_{t,j} \frac{1}{\sqrt{i(1+\eta_t \hat{\underline{L}}_{t,i})}} = \sum_j q_{t,j} q_{t,i}.$$

On the other hand, when $\frac{1}{(1+\eta_t \hat{\underline{L}}_{t,i})} \geq \frac{1}{i^{1/\alpha}}$, we have

$$\frac{1}{p_{t,i} i^{1+1/\alpha}} = \sum_j q_{t,j} \frac{\sqrt{i^{1+1/\alpha}}}{i^{1+1/\alpha}} = \sum_j q_{t,j} \frac{1}{\sqrt{i^{1+1/\alpha}}} = \sum_j q_{t,j} q_{t,i}.$$

Therefore, in any cases, we obtain

$$\mathbb{E}\left[ \hat{\ell}_{t,i} \left( \phi_i(\eta_t \hat{L}_t) - \phi_i(\eta_t \hat{L}_{t+1}) \right) \Big| \hat{L}_t \right] \leq e(\alpha+1)\eta_t \sum_{j \in [K]} q_{t,j} q_{t,i},$$

which concludes the proof. $\qquad\square$

### A.3 PROOF FOR THE PENALTY TERM (LEMMA 6)

**Lemma 6 (restated)** *For any $t \in [T]$, Algorithm 1 with $\alpha > 1$ satisfies*

$$\mathbb{E}_{r_t \sim \mathcal{P}_\alpha^K}\left[ r_{t,i_t} - r_{t,i^*} \Big| \hat{L}_t \right] \leq \frac{\alpha}{\alpha-1} \sum_{i \neq i^*} \frac{1}{(1+\eta_t \hat{\underline{L}}_{t,i})^{\alpha-1}} \wedge C_\alpha K^{\frac{1}{\alpha}},$$

*where $C_\alpha = \frac{2\alpha^3 + (e-2)\alpha^2}{(\alpha-1)(2\alpha-1)}$.*

*Proof.* The proof of this lemma is almost the same as that of Lemma 12 of Lee et al. (2024), except that we provide a slightly tighter bound.

By the choice of Pareto perturbations, we have

$$\mathbb{E}\left[ r_{t,i_t} - r_{t,i^*} \Big| \hat{L}_t \right] \leq \sum_{i \neq i^*} \mathbb{E}\left[ \mathbb{1}[I_t = i] r_{t,i} \Big| \hat{L}_t \right]$$

$$= \int_1^\infty \sum_{i \neq i^*} \left( \frac{\alpha}{(z+\eta_t \hat{\underline{L}}_{t,i})^\alpha} \right) \prod_{j \neq i} \left( 1 - \frac{1}{(z+\eta_t \hat{\underline{L}}_{t,j})^\alpha} \right) \mathrm{d}z$$

$$\leq \int_1^\infty \sum_{i \neq i^*} \left( \frac{\alpha}{(z+\eta_t \hat{\underline{L}}_{t,i})^\alpha} \right) \mathrm{d}z$$

$$= \frac{\alpha}{\alpha-1} \sum_{i \neq i^*} \frac{1}{(1+\eta_t \hat{\underline{L}}_{t,i})^{\alpha-1}}.$$

Let $k_\alpha(z) = \sum_{i \in [K]} \frac{1}{(z+\eta_t \hat{\underline{L}}_{t,i})^\alpha} \in (0, \frac{K}{z^\alpha}]$. Then,

$$\int_1^\infty \sum_{i \neq i^*} \left( \frac{\alpha}{(z+\eta_t \hat{\underline{L}}_{t,i})^\alpha} \right) \prod_{j \neq i} \left( 1 - \frac{1}{(z+\eta_t \hat{\underline{L}}_{t,j})^\alpha} \right) \mathrm{d}z$$

$$\leq \int_1^\infty \sum_{i \neq i^*} \left( \frac{\alpha}{(z+\eta_t \hat{\underline{L}}_{t,i})^\alpha} \right) \exp\left( -\sum_{j \neq i} \left( 1 - \frac{1}{(z+\eta_t \hat{\underline{L}}_{t,j})^\alpha} \right) \right) \mathrm{d}z$$

$$\leq e\alpha \int_1^\infty \sum_{i \neq i^*} \left( \frac{\alpha}{(z+\eta_t \hat{\underline{L}}_{t,i})^\alpha} \right) e^{-k_\alpha(z)} \mathrm{d}z \leq e \int_1^\infty k_\alpha(z) e^{-k_\alpha(z)} \mathrm{d}z.$$

Since $xe^{-x} \leq e^{-1}$ for $x \geq 0$ and $xe^{-x}$ is increasing for $x \leq 1$ and $k_\alpha(z)$ holds for $z \geq K^{1/\alpha}$, we have

$$e\alpha \int_1^\infty k_\alpha(z)e^{-k_\alpha(z)}\mathrm{d}z \leq \alpha \int_1^{K^{1/\alpha}} 1\mathrm{d}z + e \int_{K^{1/\alpha}}^\infty \alpha \frac{K}{z^\alpha}e^{-\frac{K}{z^\alpha}}\mathrm{d}z$$

$$= \alpha(K^{1/\alpha} - 1) + eK^{1/\alpha} \int_0^1 w^{-\frac{1}{\alpha}}e^{-w}\mathrm{d}w$$

$$= K^{1/\alpha}\left(\alpha + e\gamma\left(1 - \frac{1}{\alpha}, 1\right)\right) - \alpha,$$

where $\gamma$ denotes the lower incomplete gamma function. By the same arguments in Lee et al. (2024, Appendix D.1.), it holds that

$$\gamma\left(1 - \frac{1}{\alpha}, 1\right) \leq \frac{\alpha^2(1 - e^{-1})}{(\alpha - 1)(2\alpha - 1)} + \frac{\alpha e^{-1}}{\alpha - 1} = \frac{(1 + e^{-1})\alpha^2 - e^{-1}\alpha}{(\alpha - 1)(2\alpha - 1)},$$

which implies

$$e\alpha \int_1^\infty k_\alpha(z)e^{-k_\alpha(z)}\mathrm{d}z \leq \left(\alpha + \frac{(e + 1)\alpha^2 - \alpha}{(\alpha - 1)(2\alpha - 1)}\right)K^{\frac{1}{\alpha}} - \alpha$$

$$\leq \frac{2\alpha^3 + (e - 2)\alpha^2}{(\alpha - 1)(2\alpha - 1)}K^{\frac{1}{\alpha}} - \alpha,$$

which concludes the proof. $\qquad\qquad\square$

*Remark* 7. While the additional $-\alpha$ term is not directly used in the analysis, it is easy to observe that the upper bound vanishes as $\alpha \to \infty$. This behavior is intuitive since larger values of $\alpha$ correspond to perturbation distributions with lighter right tails, increasingly concentrated around the left endpoint at 1. In the limit, as $\alpha \to \infty$, the perturbation converges to a Dirac delta function at 1, eliminating any randomness, i.e., the difference between perturbations becomes zero.

# B    REGRET BOUND FOR ADVERSARIAL BANDITS (THEOREM 1)

**Theorem 1 (restated)** *In the adversarial regime, Algorithm 1 with $\alpha > 1$ and $\eta_t = cK^{\frac{1}{\alpha} - \frac{1}{2}}/\sqrt{t}$ for $c > 0$ satisfies* $\mathrm{Reg}(T) \leq \mathcal{O}(\sqrt{KT})$.

*Proof.* From Lemma 4, we have

$$\mathrm{Reg}(T) \leq \sum_{t=1}^T \mathbb{E}\left[\left\langle \hat{\ell}_t, \phi(\eta_t \hat{L}_t) - \phi(\eta_t \hat{L}_{t+1})\right\rangle\right] + \sum_{t=1}^{T+1}\left(\frac{1}{\eta_t} - \frac{1}{\eta_{t-1}}\right)\mathbb{E}_{r_t \sim \mathcal{D}}[r_{t,i_t} - r_{t,i^*}].$$

For the stability term (the first term), we have

$$\sum_{t=1}^T \mathbb{E}\left[\left\langle \hat{\ell}_t, \phi(\eta_t \hat{L}_t) - \phi(\eta_t \hat{L}_{t+1})\right\rangle\right] = \mathbb{E}\left[\sum_{t=1}^T \sum_{i \in [K]} \mathbb{E}\left[\hat{\ell}_{t,i}\left(\phi_i(\eta_t \hat{L}_t) - \phi_i(\eta_t \hat{L}_{t+1})\right)\Big|\hat{L}_t\right]\right]$$

$$\leq \mathbb{E}\left[\sum_{t=1}^T e(\alpha + 1)\eta_t \sum_{j \in [K]} q_{t,j} \sum_{i \in [K]} q_{t,i}\right] \quad \text{(by Lemma 5)}$$

$$\leq \sum_{t=1}^T \frac{4\alpha^2(\alpha + 1)e}{(\alpha - 1)^2} \frac{cK^{\frac{1}{\alpha} - \frac{1}{2}}}{\sqrt{t}}K^{1 - \frac{1}{\alpha}} \qquad (15)$$

$$= \sum_{t=1}^T \frac{4c\alpha^2(\alpha + 1)e}{(\alpha - 1)^2}\sqrt{\frac{K}{t}}$$

$$\leq \frac{8c\alpha^2(\alpha + 1)e}{(\alpha - 1)^2}\sqrt{KT}, \qquad (16)$$

where (15) follows by the definition of $q_{t,i}$,

$$\sum_{i \in [K]} q_{t,i} = \sum_{i \in [K]} \left( \frac{1}{1 + \eta_t \hat{\underline{L}}_{t,i}} \wedge \frac{1}{\sigma_i^{1/\alpha}} \right)^{\frac{\alpha+1}{2}} \leq \sum_{i \in [K]} i^{-\frac{1}{2} - \frac{1}{2\alpha}} \leq \frac{2\alpha}{\alpha - 1} K^{\frac{1}{2} - \frac{1}{2\alpha}}. \qquad (17)$$

For the penalty term (the second term), we have

$$\sum_{t=1}^{T+1} \left( \frac{1}{\eta_t} - \frac{1}{\eta_{t-1}} \right) \mathbb{E}_{r_t \sim \mathcal{P}_\alpha^K}[r_{t,i_t} - r_{t,i^*}]$$

$$= \frac{\mathbb{E}_{r_t \sim \mathcal{P}_\alpha^K}[r_{t,i_t} - r_{t,i^*}]}{\eta_1} + \sum_{t=2}^{T+1} \left( \frac{1}{\eta_t} - \frac{1}{\eta_{t-1}} \right) \mathbb{E}_{r_t \sim \mathcal{P}_\alpha^K}[r_{t,i_t} - r_{t,i^*}]$$

$$\leq \frac{\alpha \Gamma(1 - 1/\alpha)}{\alpha - 1} \frac{\sqrt{K}}{c} + \sum_{t=2}^{T+1} \left( \frac{1}{\eta_t} - \frac{1}{\eta_{t-1}} \right) \mathbb{E}_{r_t \sim \mathcal{P}_\alpha^K}[r_{t,i_t} - r_{t,i^*}], \qquad (18)$$

where the inequality follows from Lemma 18 of Lee et al. (2024). For the second term in (18), we have

$$\sum_{t=2}^{T+1} \left( \frac{1}{\eta_t} - \frac{1}{\eta_{t-1}} \right) \mathbb{E}\left[ \mathbb{E}_{r_t \sim \mathcal{P}_\alpha^K}\left[ r_{t,i_t} - r_{t,i^*} \Big| \hat{L}_t \right] \right] \leq \frac{K^{\frac{1}{2} - \frac{1}{\alpha}}}{c} \sum_{t=2}^{T+1} \left( \sqrt{t} - \sqrt{t-1} \right) C_\alpha K^{\frac{1}{\alpha}}$$

$$= \frac{C_\alpha \sqrt{K}}{c} \left( \sqrt{T+1} - 1 \right)$$

$$\leq \frac{C_\alpha}{c} \sqrt{KT}, \qquad (19)$$

where the last inequality follows from $\sqrt{x+1} - 1 \leq \sqrt{x}$ for $x > 0$. Therefore, from (16), (18), and (19), we obtain

$$\text{Reg}(T) \leq \left( \frac{8c\alpha^2(\alpha+1)e}{(\alpha-1)^2} + \frac{2\alpha^3 + (e-2)\alpha^2}{c(\alpha-1)(2\alpha-1)} \right) \sqrt{KT} + \frac{\alpha \Gamma(1 - 1/\alpha)}{\alpha - 1} \frac{\sqrt{K}}{c},$$

which concludes the proof. $\qquad \square$

## C REGRET BOUND FOR STOCHASTIC BANDITS (THEOREM 2)

To analyze the regret in the stochastic regime, we define the event

$$D_t := \left\{ \sum_{i \neq i^*} \frac{1}{(2^{1/\alpha} + \eta_t \hat{\underline{L}}_{t,i})^\alpha} \leq \frac{1}{2} \right\}.$$

When $D_t$ occurs, it implies that $\hat{\underline{L}}_{t,i^*} = 0$, indicating that the optimal arm $i^*$ has been accurately identified based on the information so far. In the subsequent proof, we separately analyze the cases where $D_t$ holds and where its complement $D_t^c$ holds.

**Theorem 2 (restated)** *In the stochastically constrained adversarial regime with a unique best arm $i^*$, Algorithm 1 with $\alpha \in (1, 3]$ and $\eta_t = cK^{\frac{1}{\alpha} - \frac{1}{2}}/\sqrt{t}$ for $c > 0$ satisfies*

$$\text{Reg}(T) \leq \mathcal{O}\left( \left( \sum_{t=1}^{T} \sum_{i \neq i^*} \sqrt{K} \Delta_i^{1-\alpha} t^{-\frac{\alpha}{2}} \right) + \frac{K}{\Delta_{\min}} \right).$$

*Proof.* We bound the stability term and penalty terms by separately analyzing the contributions on the events $D_t$ and $D_t^c$.

**Stability term** For the stability term, we start from

$$\sum_{t=1}^{T} \mathbb{E}\left[\left\langle \hat{\ell}_t, \phi(\eta_t \hat{L}_t) - \phi(\eta_t \hat{L}_{t+1})\right\rangle\right] = \mathbb{E}\left[\sum_{t=1}^{T} \sum_{i \in [K]} \mathbb{E}\left[\hat{\ell}_{t,i}\left(\phi_i(\eta_t \hat{L}_t) - \phi_i(\eta_t \hat{L}_{t+1})\right)\Big|\hat{L}_t\right]\right]. \quad (20)$$

On $D_t$, we separate the contribution of the optimal arm $i^*$ from that of the suboptimal arms for a tighter analysis. For the suboptimal arms, Lemma 5 yields

$$\sum_{t=1}^{T} \mathbb{1}[D_t] \sum_{i \neq i^*} \mathbb{E}\left[\hat{\ell}_{t,i}\left(\phi_i(\eta_t \hat{L}_t) - \phi_i(\eta_t \hat{L}_{t+1})\right)\Big|\hat{L}_t\right] = \sum_{t=1}^{T} \mathbb{1}[D_t] \frac{cK^{\frac{1}{\alpha} - \frac{1}{2}}e(\alpha+1)}{\sqrt{t}} \sum_{j \in [K]} q_{t,j} \sum_{i \neq i^*} q_{t,i}$$

$$\leq \sum_{t=1}^{T} \mathbb{1}[D_t] \frac{cK^{\frac{1}{\alpha} - \frac{1}{2}}e(\alpha+1)}{\sqrt{t}} \sum_{j \in [K]} q_{t,j} \sum_{i \neq i^*} (2e^2 w_{t,i})^{1 - \frac{1}{\alpha}}.$$

Here, the last step follows from

$$q_{t,i} \leq \left(\frac{1}{1 + \eta_t \underline{\hat{L}}_{t,i}}\right)^{\frac{\alpha+1}{2}} \leq (2e^2 w_{t,i})^{\frac{1}{2} + \frac{1}{2\alpha}} \leq (2e^2 w_{t,i})^{1 - \frac{1}{\alpha}}, \quad (21)$$

where the second inequality follows from Lemma 10, and the last one holds since $1 - \frac{1}{\alpha} \leq \frac{1}{2} + \frac{1}{2\alpha}$ for $\alpha \in (1, 3]$. For the optimal arm, Lemma 9 gives

$$\sum_{t=1}^{T} \mathbb{1}[D_t] \mathbb{E}\left[\hat{\ell}_{t,i^*}\left(\phi_{i^*}(\eta_t \hat{L}_t) - \phi_{i^*}(\eta_t \hat{L}_{t+1})\right)\Big|\hat{L}_t\right]$$

$$\leq \sum_{t=1}^{T} \mathbb{1}[D_t] \left[\sum_{j \in [K]} q_{t,j} \sum_{i \neq i^*} \frac{e(1 - e^{-1})\eta_t \alpha}{(1 - \zeta)^{\alpha+1}(1 + \eta_t \underline{\hat{L}}_{t,i})^{\alpha+1}} + \frac{1}{1 - e^{-1}}(1 - e^{-1})^{\frac{\zeta}{\eta_t}}\left(\frac{\zeta}{\eta_t} + e\right)\right]$$

$$\leq \sum_{t=1}^{T} \mathbb{1}[D_t] \frac{cK^{\frac{1}{\alpha} - \frac{1}{2}}e(1 - e^{-1})\alpha}{(1 - \zeta)^{\alpha+1}\sqrt{t}} \sum_{j \in [K]} q_{t,j} \sum_{i \neq i^*} \left(\frac{1}{1 + \eta_t \underline{\hat{L}}_{t,i}}\right)^{\frac{\alpha+1}{2}} + \mathcal{O}\left(c^2 K^{\frac{2}{\alpha} - 1}\right) \quad \text{(by (37))}$$

$$\leq \sum_{t=1}^{T} \mathbb{1}[D_t] \frac{cK^{\frac{1}{\alpha} - \frac{1}{2}}e(1 - e^{-1})\alpha}{(1 - \zeta)^{\alpha+1}\sqrt{t}} \sum_{j \in [K]} q_{t,j} \sum_{i \neq i^*} (2e^2 w_{t,i})^{1 - \frac{1}{\alpha}} + \mathcal{O}\left(c^2 K^{\frac{2}{\alpha} - 1}\right). \quad \text{(by (21))}$$

Combining the contributions from both the optimal and suboptimal arms, we obtain

$$\sum_{t=1}^{T} \mathbb{1}[D_t] \sum_{i \in [K]} \mathbb{E}\left[\hat{\ell}_{t,i}\left(\phi_i(\eta_t \hat{L}_t) - \phi_i(\eta_t \hat{L}_{t+1})\right)\Big|\hat{L}_t\right]$$

$$= \sum_{t=1}^{T} \mathbb{1}[D_t]\left(\alpha + 1 + \frac{\alpha(1 - e^{-1})}{(1 - \zeta)^{\alpha+1}}\right) \frac{cK^{\frac{1}{\alpha} - \frac{1}{2}}e}{\sqrt{t}} \sum_{j \in [K]} q_{t,j} \sum_{i \neq i^*} (2e^2 w_{t,i})^{1 - \frac{1}{\alpha}} + \mathcal{O}\left(c^2 K^{\frac{2}{\alpha} - 1}\right)$$

$$\leq \sum_{t=1}^{T} \mathbb{1}[D_t]\left(\frac{2c\alpha(\alpha+1)e}{\alpha - 1} + \frac{2c\alpha^2(1 - e^{-1})e}{(\alpha - 1)(1 - \zeta)^{\alpha+1}}\right) \frac{K^{\frac{1}{2\alpha}}}{\sqrt{t}} \sum_{i \neq i^*} (2e^2 w_{t,i})^{1 - \frac{1}{\alpha}} + \mathcal{O}\left(c^2 K^{\frac{2}{\alpha} - 1}\right),$$

where the last step follows from (17). On $D_t^c$, we have

$$\sum_{t=1}^{T} \mathbb{1}[D_t^c] \mathbb{E}\left[\left\langle \hat{\ell}_t, \phi(\eta_t \hat{L}_t) - \phi(\eta_t \hat{L}_{t+1})\right\rangle\Big|\hat{L}_t\right] = \sum_{t=1}^{T} \mathbb{1}[D_t^c] \frac{cK^{\frac{1}{\alpha} - \frac{1}{2}}e(\alpha+1)}{\sqrt{t}} \sum_{j \in [K]} q_{t,j} \sum_{i \in [K]} q_{t,i}$$

$$\leq \sum_{t=1}^{T} \mathbb{1}[D_t^c] \frac{4\alpha^2(\alpha+1)e}{(\alpha - 1)^2} \frac{cK^{\frac{1}{\alpha} - \frac{1}{2}}}{\sqrt{t}} K^{1 - \frac{1}{\alpha}}$$

$$\text{(by (17))}$$

$$= \sum_{t=1}^{T} \mathbb{1}[D_t^c] \frac{4c\alpha^2(\alpha+1)e}{(\alpha - 1)^2} \sqrt{\frac{K}{t}}.$$

Combining the bounds for both $D_t$ and $D_t^c$, the stability term can be bounded as

$$
\sum_{t=1}^{T} \mathbb{E}\Big[\Big\langle \hat{\ell}_t, \phi(\eta_t \hat{L}_t) - \phi(\eta_t \hat{L}_{t+1}) \Big\rangle\Big]
$$

$$
\leq \mathbb{E}\left[\sum_{t=1}^{T} \mathbb{1}[D_t]\left(\frac{2c\alpha(\alpha+1)e}{\alpha-1} + \frac{2c\alpha^2(1-e^{-1})e}{(\alpha-1)(1-\zeta)^{\alpha+1}}\right)\frac{K^{\frac{1}{2\alpha}}}{\sqrt{t}}\sum_{i\neq i^*}(2e^2 w_{t,i})^{1-\frac{1}{\alpha}}\right]
$$

$$
+ \mathbb{E}\left[\sum_{t=1}^{T} \mathbb{1}[D_t^c]\frac{4c\alpha^2(\alpha+1)e}{(\alpha-1)^2}\sqrt{\frac{K}{t}}\right] + \mathcal{O}\Big(c^2 K^{\frac{2}{\alpha}-1}\Big). \tag{22}
$$

**Penalty term** For the penalty term, we can start from (18):

$$
\sum_{t=1}^{T+1}\left(\frac{1}{\eta_t} - \frac{1}{\eta_{t-1}}\right)\mathbb{E}_{r_t\sim\mathcal{P}_\alpha^K}[r_{t,i_t} - r_{t,i^*}]
$$

$$
\leq \sum_{t=2}^{T+1}\left(\frac{1}{\eta_t} - \frac{1}{\eta_{t-1}}\right)\mathbb{E}_{r_t\sim\mathcal{P}_\alpha^K}[r_{t,i_t} - r_{t,i^*}] + \frac{\alpha\Gamma(1-1/\alpha)}{\alpha-1}\frac{\sqrt{K}}{c}. \tag{23}
$$

On $D_t$, we obtain

$$
\sum_{t=2}^{T+1} \mathbb{1}[D_t]\left(\frac{1}{\eta_t} - \frac{1}{\eta_{t-1}}\right)\mathbb{E}_{r_t\sim\mathcal{P}_\alpha^K}\Big[r_{t,i_t} - r_{t,i^*}\,\Big|\,\hat{L}_t\Big]
$$

$$
\leq \sum_{t=2}^{T+1} \mathbb{1}[D_t]\frac{\alpha}{\alpha-1}\left(\frac{1}{\eta_t} - \frac{1}{\eta_{t-1}}\right)\sum_{i\neq i^*}\frac{1}{(1+\eta_t\underline{\hat{L}}_{t,i})^{\alpha-1}}. \qquad \text{(by Lemma 6)}
$$

For $t \geq 2$, Lemma 10 implies

$$
\left(\frac{1}{\eta_t} - \frac{1}{\eta_{t-1}}\right)\sum_{i\neq i^*}\frac{1}{(1+\eta_t\underline{\hat{L}}_{t,i})^{\alpha-1}} \leq \frac{K^{\frac{1}{2}-\frac{1}{\alpha}}}{2c\sqrt{t-1}}\sum_{i\neq i^*}(2e^2 w_{t,i})^{1-\frac{1}{\alpha}},
$$

which yields

$$
\sum_{t=2}^{T+1} \mathbb{1}[D_t]\left(\frac{1}{\eta_t} - \frac{1}{\eta_{t-1}}\right)\mathbb{E}_{r_t\sim\mathcal{P}_\alpha^K}\Big[r_{t,i_t} - r_{t,i^*}\,\Big|\,\hat{L}_t\Big] \leq \sum_{t=1}^{T} \mathbb{1}[D_t]\frac{\alpha}{2c(\alpha-1)}\frac{K^{\frac{1}{2\alpha}}}{\sqrt{t}}\sum_{i\neq i^*}(2e^2 w_{t,i})^{1-\frac{1}{\alpha}},
$$

where we used the fact that $\frac{1}{2} - \frac{1}{\alpha} \leq \frac{1}{2\alpha}$ for $\alpha \in (1,3]$.

On $D_t^c$, we obtain

$$
\sum_{t=2}^{T+1} \mathbb{1}[D_t^c]\left(\frac{1}{\eta_t} - \frac{1}{\eta_{t-1}}\right)\mathbb{E}_{r_t\sim\mathcal{P}_\alpha^K}\Big[r_{t,i_t} - r_{t,i^*}\,\Big|\,\hat{L}_t\Big] \leq \sum_{t=2}^{T} \mathbb{1}[D_t^c]\left(\frac{1}{\eta_t} - \frac{1}{\eta_{t-1}}\right)C_\alpha K^{\frac{1}{\alpha}}
$$
$$
\text{(by Lemma 6)}
$$

$$
\leq \sum_{t=1}^{T} \mathbb{1}[D_t^c]\frac{K^{\frac{1}{2}-\frac{1}{\alpha}}}{c\sqrt{2}\sqrt{t}}C_\alpha K^{\frac{1}{\alpha}}
$$

$$
= \sum_{t=1}^{T} \mathbb{1}[D_t^c]\frac{C_\alpha}{c\sqrt{2}}\sqrt{\frac{K}{t}},
$$

where the first step assumes $\eta_T = \eta_{T+1}$ for simplicity and the second step is due to the fact that $\sqrt{t/(t-1)} \leq \sqrt{2}$ for $t \geq 2$. Here, the assumption $\eta_T = \eta_{T+1}$ does not affect the behavior of the algorithm, as the procedure terminates at round $T$. Although this assumes knowledge of $T$, even without it one can just introduce an additional $\mathcal{O}(1/\sqrt{T+1})$ term, which does not affect the overall

regret for sufficiently large $T$. Combining the bounds for both $D_t$ and $D_t^c$, the penalty term can be bounded as

$$\sum_{t=1}^{T+1} \left( \frac{1}{\eta_t} - \frac{1}{\eta_{t-1}} \right) \mathbb{E}_{r_t \sim \mathcal{P}_\alpha^K} [r_{t,i_t} - r_{t,i^*}]$$

$$\leq \mathbb{E} \left[ \sum_{t=1}^{T} \mathbb{1}[D_t] \frac{\alpha}{2c(\alpha-1)} \frac{K^{\frac{1}{2\alpha}}}{\sqrt{t}} \sum_{i \neq i^*} (2e^2 w_{t,i})^{1-\frac{1}{\alpha}} + \sum_{t=1}^{T} \mathbb{1}[D_t^c] \frac{C_\alpha}{c\sqrt{2}} \sqrt{\frac{K}{t}} \right] + \frac{\alpha \Gamma(1-1/\alpha)}{\alpha-1} \frac{\sqrt{K}}{c}. \tag{24}$$

Finally, combining (22) with (24), the regret can be upper bounded as

$$\text{Reg}(T) \leq \mathbb{E} \left[ \sum_{t=1}^{T} \mathbb{1}[D_t] \left( \frac{2c\alpha(\alpha+1)e}{\alpha-1} + \frac{2c\alpha^2(1-e^{-1})e}{(\alpha-1)(1-\zeta)^{\alpha+1}} + \frac{\alpha}{2c(\alpha-1)} \right) \frac{K^{\frac{1}{2\alpha}}}{\sqrt{t}} \sum_{i \neq i^*} (2e^2 w_{t,i})^{1-\frac{1}{\alpha}} \right]$$

$$+ \mathbb{E} \left[ \sum_{t=1}^{T} \mathbb{1}[D_t^c] \left( \frac{4c\alpha^2(\alpha+1)e}{(\alpha-1)^2} + \frac{C_\alpha}{c\sqrt{2}} \right) \sqrt{\frac{K}{t}} \right] + \mathcal{O}\left( c^2 K^{\frac{2}{\alpha}-1} \right) + \frac{\alpha \Gamma(1-1/\alpha)}{\alpha-1} \frac{\sqrt{K}}{c}. \tag{25}$$

**Self-bounding technique**  We employ the self-bounding technique of Zimmert & Seldin (2021) in the stochastically constrained adversarial regime to demonstrate that our policy adapts to broader settings beyond the purely stochastic regime. Specifically,

$$\text{Reg}(T) = 2 \cdot \text{Reg}(T) - \mathbb{E} \left[ \sum_{t=1}^{T} \sum_{i \neq i^*} \Delta_i w_{t,i} \right] \qquad \text{(by (2))}$$

$$= 2 \cdot \text{Reg}(T) - \mathbb{E} \left[ \sum_{t=1}^{T} \left( \mathbb{I}[D_t] \sum_{i \neq i^*} \Delta_i w_{t,i} + \mathbb{I}[D_t^c] \sum_{i \neq i^*} \Delta_i w_{t,i} \right) \right]$$

$$\leq \mathcal{O}\left( c^2 K^{\frac{2}{\alpha}-1} \right) + \frac{2\alpha \Gamma(1-1/\alpha)}{\alpha-1} \frac{\sqrt{K}}{c} + \mathbb{E} \left[ \sum_{t=1}^{T} \mathbb{I}[D_t] \sum_{i \neq i^*} \left( \frac{Z_1(\alpha) w_{t,i}^{1-\frac{1}{\alpha}}}{\sqrt{t}} - \Delta_i w_{t,i} \right) \right]$$

$$+ \mathbb{E} \left[ \sum_{t=1}^{T} \mathbb{I}[D_t^c] \left( \left( \frac{8c\alpha^2(\alpha+1)e}{(\alpha-1)^2} + \frac{C_\alpha \sqrt{2}}{c} \right) \sqrt{\frac{K}{t}} - \frac{1-e^{-1/2}}{2} \Delta_{\min} \right) \right] \qquad \text{(by (11))}$$

$$\leq \mathcal{O}\left( c^2 K^{\frac{2}{\alpha}-1} \right) + \frac{2\alpha \Gamma(1-1/\alpha)}{\alpha-1} \frac{\sqrt{K}}{c} + \mathbb{E} \left[ \sum_{t=1}^{T} \mathbb{1}[D_t] \sum_{i \neq i^*} Z_2(\alpha) \Delta_i^{1-\alpha} t^{-\frac{\alpha}{2}} \right]$$

$$+ \mathbb{E} \left[ \sum_{t=1}^{T} \mathbb{1}[D_t^c] \max \left( \left( \frac{8c\alpha^2(\alpha+1)e}{(\alpha-1)^2} + \frac{C_\alpha \sqrt{2}}{c} \right) \sqrt{\frac{K}{t}} - \frac{1-e^{-1/2}}{2} \Delta_{\min}, 0 \right) \right], \tag{26}$$

where the last step follows from

$$\sum_{i \neq i^*} \left( \frac{Z_1(\alpha) w_{t,i}^{1-\frac{1}{\alpha}}}{\sqrt{t}} - \Delta_i w_{t,i} \right) \leq \sum_{i \neq i^*} \max_{w \in [0,1]} \left( \frac{Z_1(\alpha) w^{1-\frac{1}{\alpha}}}{\sqrt{t}} - \Delta_i w \right) \leq \sum_{i \neq i^*} Z_2(\alpha) \Delta_i^{1-\alpha} t^{-\frac{\alpha}{2}},$$

which is a direct application of Lemma 8 in Rouyer & Seldin (2020). Here, the constants $Z_1(\alpha; \zeta)$ and $Z_2(\alpha; \zeta)$ are

$$Z_1(\alpha; \zeta) = Z_1(\alpha) = (2e^2)^{1-\frac{1}{\alpha}} \left( \frac{4c\alpha(\alpha+1)e}{\alpha-1} + \frac{4c\alpha^2(1-e^{-1})e}{(\alpha-1)(1-\zeta)^{\alpha+1}} + \frac{\alpha}{c(\alpha-1)} \right) K^{\frac{1}{2\alpha}} \tag{27}$$

and

$$Z_2(\alpha; \zeta) = Z_2(\alpha) = Z_1(\alpha)^\alpha \left( \left( \frac{\alpha-1}{\alpha} \right)^{\alpha-1} - \left( \frac{\alpha-1}{\alpha} \right)^\alpha \right). \tag{28}$$

Next, we define the time step $T_{\text{cut}}$ such that for $t > \lfloor T_{\text{cut}} \rfloor$, the last term in (26) evaluates to zero. Hence, we can bound the sum as

$$\sum_{t=1}^{T} \max\left( A\sqrt{\frac{K}{t}} - B \cdot \Delta_{\min}, 0 \right) \le \sum_{t=1}^{\lfloor T_{\text{cut}} \rfloor} \left( A\sqrt{\frac{K}{t}} - B \cdot \Delta_{\min} \right) \le \frac{2A^2}{B} \frac{K}{\Delta_{\min}}, \qquad (29)$$

where $T_{\text{cut}} := \frac{A^2 K}{B^2 \Delta_{\min}^2}$. Using this, we can upper bound (26) as

$$\mathcal{O}\left( c^2 K^{\frac{2}{\alpha}-1} \right) + \frac{2\alpha\Gamma(1-1/\alpha)}{\alpha-1} \frac{\sqrt{K}}{c} + \mathbb{E}\left[ \sum_{t=1}^{T} \mathbb{1}[D_t] \sum_{i\neq i^*} Z_2(\alpha)\Delta_i^{1-\alpha}t^{-\frac{\alpha}{2}} \right]$$

$$+ \mathbb{E}\left[ \sum_{t=1}^{T} \mathbb{1}[D_t^c] \max\left( \left( \frac{8c\alpha^2(\alpha+1)e}{(\alpha-1)^2} + \frac{C_\alpha\sqrt{2}}{c} \right)\sqrt{\frac{K}{t}} - \frac{1-e^{-1/2}}{2}\Delta_{\min}, 0 \right) \right]$$

$$\le \mathcal{O}\left( c^2 K^{\frac{2}{\alpha}-1} \right) + \frac{2\alpha\Gamma(1-1/\alpha)}{\alpha-1} \frac{\sqrt{K}}{c} + \sum_{t=1}^{T} \sum_{i\neq i^*} Z_2(\alpha)\Delta_i^{1-\alpha}t^{-\frac{\alpha}{2}}$$

$$+ \frac{4}{1-e^{-1/2}} \left( \frac{8c\alpha^2(\alpha+1)e}{(\alpha-1)^2} + \frac{C_\alpha\sqrt{2}}{c} \right)^2 \frac{K}{\Delta_{\min}}, \qquad \text{(by (29))}$$

which implies

$$\text{Reg}(T) \le \mathcal{O}\left( \left( \sum_{t=1}^{T} \sum_{i\neq i^*} \sqrt{K}\Delta_i^{1-\alpha}t^{-\frac{\alpha}{2}} \right) + \frac{K}{\Delta_{\min}} \right),$$

and thus concludes the proof. □

**Corollary 3 (restated)** *In the stochastically constrained adversarial regime with a unique best arm $i^*$, Algorithm 1 with $\alpha = 3$ and $\eta_t = cK^{\frac{1}{\alpha}-\frac{1}{2}}/\sqrt{t}$ for $c > 0$ satisfies*

$$\text{Reg}(T) \le \mathcal{O}\left( \sqrt{\frac{K}{\Delta_{\min}} \sum_{i\neq i^*} \frac{1}{\Delta_i}} + \frac{K}{\Delta_{\min}} \right).$$

*Proof.* By Theorem 2, for $\alpha \in (1, 3]$, the regret of our policy satisfies

$$\text{Reg}(T) \le \mathcal{O}\left( \left( \sum_{t=1}^{T} \sum_{i\neq i^*} \sqrt{K}\Delta_i^{1-\alpha}t^{-\frac{\alpha}{2}} \right) + \frac{K}{\Delta_{\min}} \right),$$

which depends on the time horizon $T$. Inspired by Theorem 4 of Rouyer & Seldin (2020), we next derive a $T$-independent bound for $\alpha \in (2, 3]$ and identify the value of $\alpha$ that minimizes this bound.

Let $T_0 := D\left( \frac{x}{\Delta_{\min}} \right)^2$ for $D \ge 1$ and $x \in \mathbb{R}_{>0}$. For $t \le T_0$, the proof follows identically to the adversarial regime, yielding a contribution of order $\mathcal{O}(\sqrt{KT_0})$. For the remaining rounds $t > T_0$, we apply the same argument as in the stochastic case, i.e., (25), which gives

$$\text{Reg}(T) \le \mathbb{E}\left[ \sum_{t=T_0+1}^{T} \mathbb{1}[D_t]\left( \frac{2c\alpha(\alpha+1)e}{\alpha-1} + \frac{2c\alpha^2(1-e^{-1})e}{(\alpha-1)(1-\zeta)^{\alpha+1}} + \frac{\alpha}{2c(\alpha-1)} \right) \frac{K^{\frac{1}{2\alpha}}}{\sqrt{t}} \sum_{i\neq i^*} (2e^2 w_{t,i})^{1-\frac{1}{\alpha}} \right]$$

$$+ \mathbb{E}\left[ \sum_{t=T_0+1}^{T} \mathbb{1}[D_t^c]\left( \frac{4c\alpha^2(\alpha+1)e}{(\alpha-1)^2} + \frac{C_\alpha}{c\sqrt{2}} \right)\sqrt{\frac{K}{t}} \right] + \left( \frac{8c\alpha^2(\alpha+1)e}{(\alpha-1)^2} + \frac{C_\alpha}{c} \right)\sqrt{KT_0}$$

$$+ \mathcal{O}\left( c^2 K^{\frac{2}{\alpha}-1} \right) + \frac{\alpha\Gamma(1-1/\alpha)}{\alpha-1} \frac{\sqrt{K}}{c}, \qquad (30)$$

for any $\zeta \in (0,1)$ and $\alpha \in (1,3]$. Here, $C_\alpha = \frac{2\alpha^3 + (e-2)\alpha^2}{(\alpha-1)(2\alpha-1)}$ denotes the constant defined in Lemma 6. Following the similar steps as Theorem 2, we obtain

$$
\mathrm{Reg}(T) = 2 \cdot \mathrm{Reg}(T) - \mathbb{E}\left[\sum_{t=1}^{T}\sum_{i \neq i^*} \Delta_i w_{t,i}\right] \tag{31}
$$

$$
\leq 2 \cdot \mathrm{Reg}(T) - \mathbb{E}\left[\sum_{t=T_0+1}^{T}\sum_{i \neq i^*} \Delta_i w_{t,i}\right]
$$

$$
\leq \mathbb{E}\left[\sum_{t=T_0+1}^{T} \mathbb{1}[D_t] \sum_{i \neq i^*} \left(\frac{Z_1(\alpha) w_{t,i}^{1-\frac{1}{\alpha}}}{\sqrt{t}} - \Delta_i w_{t,i}\right)\right]
$$

$$
+ \mathbb{E}\left[\sum_{t=T_0+1}^{T} \mathbb{1}[D_t^c]\left(\left(\frac{8c\alpha^2(\alpha+1)e}{(\alpha-1)^2} + \frac{C_\alpha\sqrt{2}}{c}\right)\sqrt{\frac{K}{t}} - \frac{1-e^{-1/2}}{2}\Delta_{\min}\right)\right]
$$

$$
+ \left(\frac{16c\alpha^2(\alpha+1)e}{(\alpha-1)^2} + \frac{2C_\alpha}{c}\right)\sqrt{KT_0} + \mathcal{O}\left(c^2 K^{\frac{2}{\alpha}-1}\right) + \frac{2\alpha\Gamma(1-1/\alpha)}{\alpha-1}\frac{\sqrt{K}}{c}
$$

$$
\leq \sum_{t=T_0+1}^{T}\sum_{i \neq i^*} Z_2(\alpha)\Delta_i^{1-\alpha} t^{-\frac{\alpha}{2}} + \frac{4}{1-e^{-1/2}}\left(\frac{8c\alpha^2(\alpha+1)e}{(\alpha-1)^2} + \frac{C_\alpha\sqrt{2}}{c}\right)^2 \frac{K}{\Delta_{\min}}
$$

$$
+ \left(\frac{16c\alpha^2(\alpha+1)e}{(\alpha-1)^2} + \frac{2C_\alpha}{c}\right)\sqrt{KT_0} + \mathcal{O}\left(c^2 K^{\frac{2}{\alpha}-1}\right) + \frac{2\alpha\Gamma(1-1/\alpha)}{\alpha-1}\frac{\sqrt{K}}{c}. \tag{32}
$$

Here, $Z_1(\alpha)$ and $Z_2(\alpha)$ are the constants defined in (27) and (28), respectively. By Lemma 12, for $\alpha \in (2,3]$, the first term in (32) can be bounded as

$$
\sum_{t=T_0+1}^{T}\sum_{i \neq i^*} Z_2(\alpha)\Delta_i^{1-\alpha} t^{-\frac{\alpha}{2}} \leq \sum_{i \neq i^*} Z_3(\alpha)\frac{\sqrt{K}D^{1-\frac{\alpha}{2}}}{\Delta_i},
$$

which gives

$$
\mathrm{Reg}(T) \leq \sum_{i \neq i^*} Z_3(\alpha)\frac{\sqrt{K}D^{1-\frac{\alpha}{2}}}{\Delta_i} + \frac{4}{1-e^{-1/2}}\left(\frac{8c\alpha^2(\alpha+1)e}{(\alpha-1)^2} + \frac{C_\alpha\sqrt{2}}{c}\right)^2 \frac{K}{\Delta_{\min}}
$$

$$
+ \left(\frac{16c\alpha^2(\alpha+1)e}{(\alpha-1)^2} + \frac{2C_\alpha}{c}\right)\sqrt{KT_0} + \mathcal{O}\left(c^2 K^{\frac{2}{\alpha}-1}\right) + \frac{2\alpha\Gamma(1-1/\alpha)}{\alpha-1}\frac{\sqrt{K}}{c}
$$

$$
= \sum_{i \neq i^*} Z_3(\alpha)\frac{\sqrt{K}D^{1-\frac{\alpha}{2}}}{\Delta_i} + \frac{4}{1-e^{-1/2}}\left(\frac{8c\alpha^2(\alpha+1)e}{(\alpha-1)^2} + \frac{C_\alpha\sqrt{2}}{c}\right)^2 \frac{K}{\Delta_{\min}}
$$

$$
+ \left(\frac{16c\alpha^2(\alpha+1)e}{(\alpha-1)^2} + \frac{2C_\alpha}{c}\right)\frac{x\sqrt{KD}}{\Delta_{\min}} + \mathcal{O}\left(c^2 K^{\frac{2}{\alpha}-1}\right) + \frac{2\alpha\Gamma(1-1/\alpha)}{\alpha-1}\frac{\sqrt{K}}{c}, \tag{33}
$$

where

$$
Z_3(\alpha) = \frac{2x^{2-\alpha}Z_2(\alpha)}{(\alpha-2)\sqrt{K}}.
$$

The first and third terms in (33) are minimized when $\alpha = 3$ and

$$
D = \frac{Z_3(3)}{x(144ce + 2C_3/c)}\sum_{i \neq i^*}\frac{\Delta_{\min}}{\Delta_i},
$$

which, by the AM-GM inequality, leads to the bound

$$\sum_{i \neq i^*} \frac{Z_3(3)}{\Delta_i} \frac{\sqrt{K}}{\sqrt{D}} + \left(144ce + \frac{2C_3}{c}\right) \frac{x\sqrt{KD}}{\Delta_{\min}} \leq 2\sqrt{\left(144ce + \frac{2C_3}{c}\right) \frac{xZ_3(3)K}{\Delta_{\min}} \sum_{i \neq i^*} \frac{1}{\Delta_i}} \quad (34)$$

$$\leq \mathcal{O}\left(\sqrt{\frac{K}{\Delta_{\min}} \sum_{i \neq i^*} \frac{1}{\Delta_i}}\right). \quad (35)$$

Note that the feasible range of $x \in \mathbb{R}_{>0}$ is determined by the two constraints $T_0 \leq T_{\mathrm{cut}}$ and $D \geq 1$:

$$x \leq \min\left\{ \frac{4(144ce + 2C_3/c)(72ce + C_3\sqrt{2}/c)^2}{Z_3(3)(1 - e^{-1/2})^2} \frac{K}{\Delta_{\min}} \left(\sum_{i \neq i^*} \frac{1}{\Delta_i}\right)^{-1}, \frac{Z_3(3)}{144ce + 2C_3/c} \sum_{i \neq i^*} \frac{\Delta_{\min}}{\Delta_i} \right\}.$$

Any choice of $x$ within this range is valid, since $x$ cancels out in (34) through the term $xZ_3(3)$, as $Z_3(3)$ contains $x^{-1}$, and thus does not affect the final bound. Finally, from (33) and (35), we obtain

$$\mathrm{Reg}(T) \leq \mathcal{O}\left(\sqrt{\frac{K}{\Delta_{\min}} \sum_{i \neq i^*} \frac{1}{\Delta_i}} + \frac{K}{\Delta_{\min}}\right),$$

which concludes the proof.

**The choice of $c$** We determine the choice of $c$ in (34). Specifically,

$$\left(144ce + \frac{2C_3}{c}\right) xZ_3(3) = \frac{32e^4}{27}\left(144ce + \frac{36 + 9e}{5c}\right)\left(24ce + \frac{18ce(1 - e^{-1})}{(1 - \zeta)^4} + \frac{3}{2c}\right)^3. \quad (36)$$

For notational simplicity, we express the RHS in the form $f(c) = (h_1 c + h_2/c)(h_3 c + h_4/c)^3$, where $h_1, h_2, h_3, h_4$ are constants determined by the coefficients above. To minimize $f(c)$, we substitute $x = c^2$ and define $g(x) = (h_1 x + h_2)(h_3 x + h_4)^3/x^2$. Differentiating $g(x)$ with respect to $x$ gives

$$g'(x) = \frac{(h_3 x + h_4)^2 (2h_1 h_3 x^2 + (h_2 h_3 - h_1 h_4)x - 2h_2 h_4)}{x^3}.$$

Thus, the stationary points of $g(x)$, where $g'(x) = 0$, are obtained by solving the quadratic equation $2h_1 h_3 x^2 + (h_2 h_3 - h_1 h_4)x - 2h_2 h_4 = 0$. This yields

$$x^* = \frac{-h_2 h_3 + h_1 h_4 + \sqrt{(h_2 h_3 - h_1 h_4)^2 + 16h_1 h_2 h_3 h_4}}{4h_1 h_3}.$$

Recalling that $x = c^2$, the optimal choice of $c$ is given by

$$c^* = \sqrt{x^*}.$$

When $\zeta = 10^{-1}$, the above computation gives

$$c^* \simeq 0.128, \quad f(c) \simeq 25799.360.$$

As $\zeta$ decreases, the resulting constant decreases.

*Remark* 8. There are some possible ways to further reduce the leading multiplicative constant of (36) in the regret bound. Firstly, in (31), one may introduce a parameter $\kappa \in (0, 1)$ and consider $(30) - \kappa \times (2)$. In our analysis, we use $\kappa = 1/2$ for simplicity, but optimizing over $\kappa$ can reduce the constant. Secondly, in (10), one may define

$$D_t = \left\{ \sum_{i \neq i^*} \frac{1}{(y^{1/\alpha} + \eta_t \hat{\underline{L}}_{t,i})^\alpha} \leq \frac{1}{y} \right\},$$

where we set $y = 2$ for simplicity in our analysis. By optimizing $y$, the constant in the regret bound can be reduced. For instance, the term $2e^2$ in (21) contributes to the constant, and under the above definition of $D_t$, it becomes $ye^{\frac{2}{y-1}}$, which can be decreased by choosing an appropriate $y$. Finally, in the process of choosing $c$, we set $\zeta = 10^{-1}$ for simplicity. Together with the optimizations discussed above, tuning $\zeta$ could further reduce the constant in the regret bound.

$\square$

## D  AUXILIARY LEMMAS

Here, we include several lemmas, along with their proofs if necessary, that are used in the appendix.

**Lemma 9** (modified Lemma 25 of Lee et al. (2024)). *On $D_t$, for any $\zeta \in (0, 1)$, FTPL with Pareto perturbations of shape $\alpha > 1$ satisfies*

$$\mathbb{E}\left[\hat{\ell}_{t,i^*}\left(\phi_{i^*}(\eta_t \hat{L}_t) - \phi_{i^*}(\eta_t \hat{L}_{t+1})\right)\Big|\hat{L}_t\right]$$

$$\leq \sum_{j \in [K]} q_{t,j} \cdot \sum_{i \neq i^*} \frac{e(1 - e^{-1})\eta_t \alpha}{(1 - \zeta)^{\alpha+1}(1 + \eta_t \underline{\hat{L}}_{t,i})^{\alpha+1}} + \frac{1}{1 - e^{-1}}(1 - e^{-1})^{\frac{\zeta}{\eta_t}}\left(\frac{\zeta}{\eta_t} + e\right)$$

*and when $\eta_t = cK^{\frac{1}{\alpha} - \frac{1}{2}}/\sqrt{t}$,*

$$\sum_{t=1}^{\infty} \frac{1}{1 - e^{-1}}(1 - e^{-1})^{\frac{\zeta}{\eta_t}}\left(\frac{\zeta}{\eta_t} + e\right) \leq \mathcal{O}\left(c^2 K^{\frac{2}{\alpha} - 1}\right). \tag{37}$$

*Proof.* Note that our formulation differs slightly from that in Lee et al. (2024), as we require a tighter result for the later use of this lemma. Nevertheless, the overall proof remains almost the same and thus we only provide details for the parts that differ.

As the proof in Honda et al. (2023) and Lee et al. (2024), we consider two cases (i) $p_{t,i^*}^{-1} \leq \zeta/\eta_t$ and (ii) $p_{t,i^*}^{-1} > \zeta/\eta_t$ separately. Notice that the case (ii) can be directly obtained by Lemma 11 in Honda et al. (2023) (or Lemma 23 in Lee et al. (2024)), which shows that

$$\mathbb{E}\left[\mathbb{1}\left[\hat{\ell}_{t,i^*} > \frac{\zeta}{\eta_t}\right]\hat{\ell}_{t,i^*}\Big|\hat{L}_t\right] \leq \frac{1}{1 - e^{-1}}(1 - e^{-1})^{\frac{\zeta}{\eta_t}}\left(\frac{\zeta}{\eta_t} + e\right).$$

When $p_{t,i}^{-1} \leq \zeta/\eta_t$, on $D_t$ (where $\hat{L}_{t,i^*} = 0$), we have

$$\phi_{i^*}(\eta_t(\hat{L}_t + xe_{i^*})) = \int_1^{\infty} f(z) \prod_{j \neq i^*} F\left(z + \eta_t(\underline{\hat{L}}_{t,j} - x)\right)\mathrm{d}z.$$

This implies that for $x \leq \frac{\zeta}{\eta_t}$

$$-\frac{\mathrm{d}}{\mathrm{d}x}\phi_{i^*}(\eta_t(\hat{L}_t + xe_{i^*}))$$

$$= \int_1^{\infty} f(z)\sum_{i \neq i^*}\left(\eta_t f\left(z + \eta_t(\underline{\hat{L}}_{t,i} - x)\right)\prod_{j \neq i,i^*}\left(1 - F\left(z + \eta_t(\underline{\hat{L}}_{t,j} - x)\right)\right)\right)\mathrm{d}z$$

$$\leq \int_1^{\infty} f(z)\sum_{i \neq i^*}\left(\eta_t f\left(z + \eta_t(\underline{\hat{L}}_{t,i} - x)\right)\exp\left(-\sum_{j \neq i,i^*}\left(1 - F\left(z + \eta_t(\underline{\hat{L}}_{t,j} - x)\right)\right)\right)\right)\mathrm{d}z$$

$$\leq e\int_1^{\infty} f(z)\sum_{i \neq i^*}\left(\eta_t f\left(z + \eta_t(\underline{\hat{L}}_{t,i} - x)\right)\exp\left(-\sum_{j \neq i,i^*}\left(1 - F\left(z + \eta_t(\underline{\hat{L}}_{t,j} - x)\right)\right) - (1 - F(z))\right)\right)\mathrm{d}z$$

$$\leq e\int_1^{\infty} f(z)\sum_{i \neq i^*}\eta_t f\left(z + \eta_t(\underline{\hat{L}}_{t,i} - x)\right)\exp(-(1 - F(z)))\mathrm{d}z$$

$$= e\int_1^{\infty} f(z)\sum_{i \neq i^*}\eta_t \frac{\alpha}{(z + \eta_t(\underline{\hat{L}}_{t,i} - x))^{\alpha+1}}\exp(-(1 - F(z)))\mathrm{d}z$$

$$\leq \sum_{i \neq i^*}\frac{e\eta_t\alpha}{(1 - \zeta)^{\alpha+1}(1 + \eta_t\underline{\hat{L}}_{t,i})^{\alpha+1}}\int_1^{\infty} f(z)\exp(-(1 - F(z)))\mathrm{d}z \tag{38}$$

$$= \sum_{i \neq i^*}\frac{e(1 - e^{-1})\eta_t\alpha}{(1 - \zeta)^{\alpha+1}(1 + \eta_t\underline{\hat{L}}_{t,i})^{\alpha+1}}, \tag{39}$$

where (38) follows from $x \leq \zeta/\eta_t$ and

$$\frac{1}{(z+a-b)} \leq \frac{1}{(1+a)(1-b)}, \quad \forall z \geq 1, b < 1, a \geq 0.$$

Therefore, we have

$$\mathbb{E}\left[\mathbb{1}[\hat{\ell}_{t,i^*} \leq \zeta/\eta_t]\hat{\ell}_{t,i^*}\left(\phi_{i^*}(\eta_t\hat{L}_t) - \phi_{i^*}(\eta_t\hat{L}_{t+1})\right)\middle|\hat{L}_t\right]$$

$$\leq \mathbb{E}\left[\mathbb{1}[\hat{\ell}_{t,i} \leq \zeta/\eta_t]\hat{\ell}_{t,i^*}^2 \sum_{i\neq i^*}\frac{e(1-e^{-1})\eta_t\alpha}{(1-\zeta)^{\alpha+1}(1+\eta_t\hat{\underline{L}}_{t,i})^{\alpha+1}}\middle|\hat{L}_t\right] \quad \text{(by (39))}$$

$$\leq \mathbb{E}\left[\frac{\ell_{t,i^*}^2}{p_{t,i^*}} \sum_{i\neq i^*}\frac{e(1-e^{-1})\eta_t\alpha}{(1-\zeta)^{\alpha+1}(1+\eta_t\hat{\underline{L}}_{t,i})^{\alpha+1}}\middle|\hat{L}_t\right]$$

$$\leq \sum_{j\in[K]}q_{t,j} \sum_{i\neq i^*}\frac{e(1-e^{-1})\eta_t\alpha}{(1-\zeta)^{\alpha+1}(1+\eta_t\hat{\underline{L}}_{t,i})^{\alpha+1}},$$

where the last inequality follows from $\ell_t \in [0,1]^K$ and $p_{t,i^*} = \frac{q_{t,i^*}}{\sum_{j\in[K]}q_{t,j}}$ with $q_{t,i^*} = 1$ on $D_t$. $\quad\square$

**Lemma 10.** *For FTPL with Pareto perturbations with shape $\alpha > 1$, it holds that*

$$w_{t,i} \leq \frac{1}{(1+\eta_t\hat{\underline{L}}_{t,i})^{\alpha}}, \forall t \in [T], i \in [K] \quad \text{and} \quad w_{t,i} \geq \frac{1}{2e^2(1+\eta_t\hat{\underline{L}}_{t,i})^{\alpha}} \text{ on } D_t, \forall i \neq i^*.$$

*In addition, the optimal arm satisfies $w_{t,i^*} \geq \frac{1}{2e}$ on $D_t$.*

*Proof.* By definition of $w_{t,i}$, for any $i \in [K]$, $t \in \mathbb{N}$ and $\hat{\underline{L}}_t \in \mathbb{R}_{\geq 0}^K$, it holds that

$$w_{t,i} = \int_1^\infty f(z+\eta_t\hat{\underline{L}}_{t,i})\prod_{j\neq i}F(z+\eta_t\hat{\underline{L}}_{t,j})\mathrm{d}z = \int_1^\infty \frac{\alpha}{(z+\eta_t\hat{\underline{L}}_{t,i})^{\alpha+1}}\prod_{j\neq i}F(z+\eta_t\hat{\underline{L}}_{t,j})\mathrm{d}z.$$

**Upper bound**   The upper bound follows directly from the definition of $w_{t,i}$:

$$\int_1^\infty \frac{\alpha}{(z+\eta_t\hat{\underline{L}}_{t,i})^{\alpha+1}}\prod_{j\neq i}F(z+\eta_t\hat{\underline{L}}_{t,j})\mathrm{d}z \leq \int_1^\infty \frac{\alpha}{(z+\eta_t\hat{\underline{L}}_{t,i})^{\alpha+1}}\mathrm{d}z \leq \frac{1}{(1+\eta_t\hat{\underline{L}}_{t,i})^{\alpha}}.$$

Note that this inequality holds for all $t$, regardless of whether the event $D_t$ occurs.

**Lower bound** Since the cumulative distribution function $F$ takes value in $[0, 1]$, on $D_t$, we obtain for the second term,

$$w_{t,i} = \int_1^\infty f(z + \eta_t \underline{\hat{L}}_{t,i}) \prod_{j \neq i} F(z + \eta_t \underline{\hat{L}}_{t,j}) \mathrm{d}z$$

$$\geq \int_1^\infty f(z + \eta_t \underline{\hat{L}}_{t,i}) \prod_{j \in [K]} F(z + \eta_t \underline{\hat{L}}_{t,j}) \mathrm{d}z$$

$$\geq \int_1^\infty f(z + \eta_t \underline{\hat{L}}_{t,i}) \exp\left( - \sum_{j \in [K]} \frac{1 - F(z + \eta_t \underline{\hat{L}}_{t,j})}{F(z + \eta_t \underline{\hat{L}}_{t,j})} \right) \mathrm{d}z \quad (\because e^{-\frac{x}{1-x}} \leq 1 - x \text{ for } x < 1)$$

$$= \int_1^\infty f(z + \eta_t \underline{\hat{L}}_{t,i}) \exp\left( - \sum_{j \neq i^*} \frac{1 - F(z + \eta_t \underline{\hat{L}}_{t,j})}{F(z + \eta_t \underline{\hat{L}}_{t,j})} \right) \exp\left( -\frac{1 - F(z)}{F(z)} \right) \mathrm{d}z$$

$$(\because \underline{\hat{L}}_{t,i^*} = 0 \text{ on } D_t)$$

$$\geq \int_{2^{1/\alpha}}^\infty f(z + \eta_t \underline{\hat{L}}_{t,i}) \exp\left( - \sum_{j \neq i^*} \frac{1 - F(z + \eta_t \underline{\hat{L}}_{t,j})}{F(z + \eta_t \underline{\hat{L}}_{t,j})} \right) \exp\left( -\frac{1 - F(z)}{F(z)} \right) \mathrm{d}z$$

$$\geq \frac{1}{e} \int_{2^{1/\alpha}}^\infty f(z + \eta_t \underline{\hat{L}}_{t,i}) \exp\left( -2 \sum_{j \neq i^*} (1 - F(z + \eta_t \underline{\hat{L}}_{t,j})) \right) \mathrm{d}z \quad (\because 2^{1/\alpha} \text{ is the median})$$

$$= \frac{1}{e} \int_{2^{1/\alpha}}^\infty f(z + \eta_t \underline{\hat{L}}_{t,i}) \exp\left( -2 \sum_{j \neq i^*} \frac{1}{(z + \eta_t \underline{\hat{L}}_{t,j})^\alpha} \right) \mathrm{d}z \quad \text{(Pareto perturbation)}$$

$$\geq \frac{1}{e^2} \int_{2^{1/\alpha}}^\infty f(z + \eta_t \underline{\hat{L}}_{t,i}) \mathrm{d}z = \frac{1}{e^2} \frac{1}{(2^{1/\alpha} + \eta_t \underline{\hat{L}}_{t,i})^\alpha}. \quad \text{(Definition of } D_t \text{ in (10))}$$

Since $\frac{(x+1)^\alpha}{(x+2^{1/\alpha})^\alpha}$ is increasing with respect to $x \geq 0$ for any $\alpha > 1$, this implies that

$$\frac{(1 + \eta_t \underline{\hat{L}}_{t,i})^\alpha}{(2^{1/\alpha} + \eta_t \underline{\hat{L}}_{t,i})^\alpha} \geq \frac{1}{2} \implies \frac{1}{2(1 + \eta_t \underline{\hat{L}}_{t,i})^\alpha} \leq \frac{1}{(2^{1/\alpha} + \eta_t \underline{\hat{L}}_{t,i})^\alpha},$$

which concludes the proof for the lower bound.

**Lower bound for optimal arm** Since $\underline{\hat{L}}_{t,i^*} = 0$ on $D_t$, we obtain that

$$w_{t,i^*} = \int_1^\infty \frac{\alpha}{z^{\alpha+1}} \prod_{j \neq i^*} F(z + \eta_t \underline{\hat{L}}_{t,j}) \mathrm{d}z$$

$$\geq \int_1^\infty \frac{\alpha}{z^{\alpha+1}} \exp\left( - \sum_{j \neq i^*} \frac{1 - F(z + \eta_t \underline{\hat{L}}_{t,j})}{F(z + \eta_t \underline{\hat{L}}_{t,j})} \right) \mathrm{d}z \quad (\because e^{-\frac{x}{1-x}} \leq 1 - x \text{ for } x < 1)$$

$$\geq \int_{2^{1/\alpha}}^\infty \frac{\alpha}{z^{\alpha+1}} \exp\left( - \sum_{j \neq i^*} \frac{1 - F(z + \eta_t \underline{\hat{L}}_{t,j})}{F(z + \eta_t \underline{\hat{L}}_{t,j})} \right) \mathrm{d}z$$

$$\geq \int_{2^{1/\alpha}}^\infty \frac{\alpha}{z^{\alpha+1}} \exp\left( -2 \sum_{j \neq i^*} (1 - F(z + \eta_t \underline{\hat{L}}_{t,j})) \right) \mathrm{d}z$$

$$\geq \frac{1}{e} \int_{2^{1/\alpha}}^\infty \frac{\alpha}{z^{\alpha+1}} \mathrm{d}z = \frac{1}{2e},$$

which concludes the proof. □

**Lemma 11.** *On $D_t^c$, $\sum_{i \neq i^*} \Delta_i w_{t,i} \geq \frac{1-e^{-1/2}}{2} \Delta_{\min}$.*

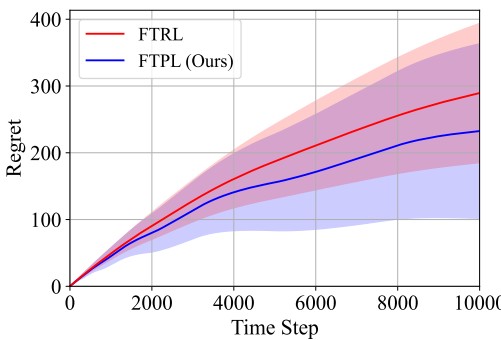
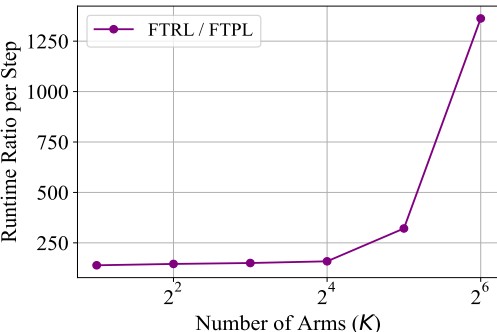

Figure 5: Adversarial regret with $\Delta = 0.0625$    Figure 6: Runtime ratio

*Proof.* By definition, we have on $D_t^c$ that

$$
w_{t,i^*} = \int_1^\infty \frac{\alpha}{(z + \eta_t \hat{\underline{L}}_{t,i^*})^{\alpha+1}} \prod_{j \neq i^*} \left(1 - \frac{1}{(z + \eta_t \hat{\underline{L}}_{t,j})^\alpha}\right) dz
$$

$$
\leq \int_1^\infty \frac{\alpha}{(z + \eta_t \hat{\underline{L}}_{t,i^*})^{\alpha+1}} \exp\left(-\sum_{j \neq i^*} \frac{1}{(z + \eta_t \hat{\underline{L}}_{t,j})^\alpha}\right) dz
$$

$$
= \int_1^{2^{1/\alpha}} \frac{\alpha}{(z + \eta_t \hat{\underline{L}}_{t,i^*})^{\alpha+1}} \exp\left(-\sum_{j \neq i^*} \frac{1}{(z + \eta_t \hat{\underline{L}}_{t,j})^\alpha}\right) dz + \int_{2^{1/\alpha}}^\infty \frac{\alpha}{(z + \eta_t \hat{\underline{L}}_{t,i^*})^{\alpha+1}} dz
$$

$$
\leq \frac{1}{\sqrt{e}} \int_1^{2^{1/\alpha}} \frac{\alpha}{(z + \eta_t \hat{\underline{L}}_{t,i^*})^{\alpha+1}} + \int_{2^{1/\alpha}}^\infty \frac{\alpha}{(z + \eta_t \hat{\underline{L}}_{t,i^*})^{\alpha+1}} dz \qquad \text{(by definition of } D_t^c\text{)}
$$

$$
\leq \frac{1}{\sqrt{e}} \int_1^{2^{1/\alpha}} \frac{\alpha}{z^{\alpha+1}} + \int_{2^{1/\alpha}}^\infty \frac{\alpha}{z^{\alpha+1}} dz = \frac{e^{-1/2}}{2} + \frac{1}{2}.
$$

Since $1 - w_{t,i^*} = \sum_{i \neq i^*} w_{t,i}$, the result follows. $\qquad \square$

**Lemma 12** (Lemma of Rouyer & Seldin (2020))**.** *Let* $T_0 := \max_{i \neq i^*} \left\lceil D\left(\frac{x}{\Delta_i}\right)^2 \right\rceil$ *for some constants* $x \in \mathbb{R}_{>0}$ *and* $D \geq 1$. *For each suboptimal arm* $i \neq i^*$, *define* $S_i(T) := \Delta_i^{1-\alpha} \sum_{t=T_0+1}^T t^{-\frac{\alpha}{2}}$. *Then* $S_i(T)$ *converges as* $T \to \infty$ *if and only if* $\alpha > 2$. *Moreover, for* $\alpha > 2$, *we have*

$$
\lim_{T \to \infty} S_i(T) \leq \frac{2}{\alpha - 2} \frac{\left(x\sqrt{D}\right)^{2-\alpha}}{\Delta_i}.
$$

# E    ADDITIONAL EXPERIMENTS

In this section, we present additional experimental results, including further experiments in the adversarial regime with $\Delta = 0.0625$, following the setup of Zimmert & Seldin (2021) described in Section 4.

Figure 5 presents the empirical performance of FTRL compared to FTPL. Our policy achieves lower cumulative regret with sublinear growth, as observed for $\Delta = 0.125$, with shaded region indicating one standard deviation.

Figure 6 shows the per-step runtime ratio of FTRL to FTPL for $K \in \{2^i : i \in [6]\}$, where the optimization step in FTRL is solved using the splitting conic solver (SCS). The $x$-axis is shown on a logarithmic scale. As the number of arms increases, the ratio grows rapidly, reaching approximately 1363 for $K = 64$. Due to the excessive runtime of FTRL for larger number of arms, the experiments were repeated only 100 times and not performed beyond $K = 64$.

