# OpenReview forum: "Follow-the-Perturbed-Leader for Decoupled Bandits: Best-of-Both-Worlds and Practicality"
_ICLR.cc/2026/Conference — ICLR 2026 Conference Withdrawn Submission_

### Official Review · Reviewer_tnbB · 2025-10-17

**Soundness:** 3
**Presentation:** 3
**Contribution:** 2
**Rating:** 4
**Confidence:** 3

**Summary:**

The paper studies the decoupled MAB problem, where an action is used for exploration while and another one is used to exploit. The authors show that a particular instantiation of the FTPL algorithm may attain best-of-both-worlds guarantees while being computationally more efficient than the state-of-the-art best-of-both-worlds algorithm for the same setting, which instead is based on FTRL.

**Strengths:**

The techniques employed in the work are interesting and the results seem both rigorous and novel. Overall, I believe that the results are well presented. Specifically, the authors did a good job in providing a precise overview of the state-of-the-arts techniques for the problem studied. Moreover, the authors put a lot of effort into explaining the main idea behind the proofs and the techniques employed. On the technical side, I believe this is a good paper.

**Weaknesses:**

The main weakness is the significance of the results obtained. Indeed, this paper focuses on a really specific topic, that is, uncoupled MAB, in which the bounds attained by FTPL do not improve the state-of-the-arts (and optimal) ones. Thus, to me, the contribution mainly lies in avoiding the convex optimization step of FTRL which I believe is not enough to meet the acceptance bar.

**Questions:**

Can the authors elaborate on the contributions of the works? I am glad to increase my score if the answers turn out to be pretty positive.

---

> ### Author Response · Authors · 2025-11-22
>
> Thank you for the time and effort you devoted to reviewing our paper. Below, we provide detailed responses to each of the weaknesses you raised.
>
> **Decoupled setting is worth studying as the achievable stochastic regret is substantially smaller**
>
> While addressing a general problem that incorporates several settings is an important and worth pursuing goal, we believe that analyzing a specific problem setting is also valuable when its achievable regret bound is substantially small/different. For example, the bandit problem with graph feedback includes the standard MAB as a special case, yet under the weakly observable regime only an $O(T^{2/3})$ regret is achievable. In the decoupled setting, the adversarial problem is as hard as the standard MAB in principle. However, the achievable stochastic regret becomes much smaller, as one can obtain a constant regret bound independent of $T$. Since the additional structure significantly reduces the attainable stochastic regret, developing efficient BOBW policies for this setting is indeed worthwhile.
>
> **Contribution even with non-SOTA analysis; computational efficiency and empirical performance of FTPL and TS**
>
> In the stochastic MAB literature, UCB and TS are the two central policies. Here, UCB and TS can be seen as a kind of (deterministic) FTRL and FTPL, respectively. While both policies achieve the same optimal performance in several settings, TS was shown to achieve a near-optimal bound (Agrawal & Goyal, 2013), which suffers larger regret than that of UCB in linear bandits (Abbasi-Yadkori et al., 2011). Nevertheless, we believe this does not degrade the contribution of analyzing TS (even if not the SOTA), as its value lies in providing meaningful theoretical understanding combined with practicality of TS: superior empirical performance, and computational efficiency. This shares the same spirit as our paper: our proposed policy shows empirically better performance in both regimes even without any computational overhead.
>
> **Road to better analysis and implication of our results for the advances of FTPL analysis**
>
> The SOTA regret bound is obtained by FTRL approach with arm-dependent learning rate and hybrid regularizer, the sum of Tsallis entropy and log-barrier (Jin et al., 2023). Therefore, the solution of convex optimization for computing $w_t$ does not have a closed formulation anymore, and thus we cannot apply computationally efficient Newton's method. In this sense, a computationally efficient FTPL policy is more attractive, where we may need to design a more sophisticated learning rate scheme such as arm-dependent learning rates. However, as we explained in the response to Reviewer XB6E (Q1-Q2), it cannot be obtained in a straightforward way as the design of sophisticated learning rate design relies on arm-selection probability $w_t$. Therefore, we believe our construction $q_t$ for FTPL, replacing the dependence on $w_t$ that FTRL requires, provides a clear conceptual and technical contribution and opens a path for deploying FTPL in settings far beyond the decoupled case.

---

### Official Review · Reviewer_XB6E · 2025-10-19

**Soundness:** 3
**Presentation:** 3
**Contribution:** 2
**Rating:** 4
**Confidence:** 4

**Summary:**

In this paper, authors study the best-of-both-worlds problem for decoupled multi-armed bandits under the Follow-the-Perturbed-Leader (FTPL) framework. Compared with previous FTRL-based algorithms which require solving a convex optimization problem at each round, the proposed algorithm is computationally efficient and obtains near-optimal results in both stochastic and adversarial regimes. Authors also conduct experiments to show that the proposed algorithm indeed outperforms Decoupled-Tsallis-INF algorithm in terms of both regret and running time.

**Strengths:**

- This paper is known as the first to study the best-of-both-worlds problem for decoupled multi-armed bandits under the Follow-the-Perturbed-Leader (FTPL) framework. The studied problem is well-motivated, given the computational issue of FTRL-based algorithms.

- The proposed algorithm is simple and intuitive. The algorithm adopts Pareto perturbations, used in previous FTPL work, to the decoupled bandit problem.

- Authors also empirically show the superiority in some experiments in terms of the regret performance and running time.

**Weaknesses:**

I have two major concerns.

- In this paper, the regret bound in the stochastic setting is $O(\sqrt{\frac{K}{\Delta_{\min} } \sum_{i \neq i^\*} \frac{1}{\Delta_i} } +\frac{K}{\Delta_{\min}})$, which is worse than the best-known result $O( \sqrt{  \sum_{i \neq i^\* } \frac{1}{\Delta_i^2} } +K )$ by [*]. For example, if the dominant term is $1/\Delta_{\min}$ (i.e., $\Delta_{\min}$ is sufficiently small) and all other arm gaps are constant level, say $\Delta_i=0.5$. In this case, the bound in this paper is dominated by $O(\frac{K}{\Delta_{\min}})$, but their bound is $O(\frac{1}{\Delta_{\min}})$. In other words, their bound is $K$ times smaller than the bound in this paper.

- The technical contribution is limited. For example, Pareto perturbation or its generalization has been widely applied in FTPL literature. While this paper is the first to use it in the decoupled multi-armed bandit problem, the techniques mostly follow previous work. The key difference that the proposed algorithm needs not to do geometric resampling is attributed to the decoupled setting, in which the algorithm can sample and weight by a self‑chosen probability distribution.

[*] Tiancheng Jin, Junyan Liu, and Haipeng Luo. Improved Best-of-Both-Worlds guarantees for multiarmed
bandits: FTRL with general regularizers and multiple optimal arms.

**Questions:**

Do authors believe that arm-dependent learning rates will improve the regret also for FTPL-based algorithms? If so, can authors provide a short and intuitive discussion on which part in your current analysis can be refined via using arm-dependent learning rates.

---

> ### Author Response · Authors · 2025-11-22
>
> Thank you for the time and effort you devoted to reviewing our paper. Below, we provide detailed responses to each of the weaknesses you raised.
>
> [W1]
>
> **Comparison to the result of Jin et al. (2023)**
>
> The regret bound obtained by Jin et al. (2023) for the stochastic setting is $O(\sqrt{\sum\_{i\ne i^\*} \frac{K}{\Delta\_i^2}}+K)$ when there exists a unique optimal arm $i^\*$. On the other hand, our current bound is $O(\sqrt{\sum\_{i\ne i^\*} \frac{K}{\Delta\_i \Delta\_{\text{min}}}} + \frac{K}{\Delta\_{\text{min}}})$. Therefore, our regret is at most a factor of $\sqrt{K}$ worse, not $K$ since $\sqrt{\sum\_{i\ne i^*} \frac{K}{\Delta\_i^2}}\geq \frac{\sqrt{K}}{\Delta\_{\text{min}}}$ holds. Moreover, when $\Delta_2 = \cdots = \Delta\_{K}$, the ratio reduces to $\sqrt{K/(K-1)}$.
>
> **Drawbacks of FTRL with adaptive learning rates**
>
> Besides the improved regret bound by Jin et al. (2023), it is important to mention that they used hybrid regularizer, the summation of Tsallis entropy and log-barrier regularization rather than naive Tsallis entropy used in Rouyer & Seldin (2020). However, such regularization comes at a computational cost: the arm-selection probabilities $w_t$ in Jin et al. (2023) lack a closed-form expression, preventing the use of Newton’s method for optimization. As a result, each iteration would require significantly more computation, as Decoupled-Tsallis-INF requires the longer times to compute $w_t$ when using convex optimization solver demonstrated in Appendix E.
>
> Given that one of the original motivations for the decoupled setting stems from applications in communication engineering, where decisions often need to be made quickly and the environment often be noisy, fast per-iteration computation with BOBW guarantee is critical. We therefore believe our approach, which preserves meaningful theoretical analysis and lightweight computation, offers a substantial practical advantage in such time-sensitive environments: a core motivation for introducing the FTPL framework to the online learning literature.
>
> [Q1-Q2]
>
> To the best of our knowledge, no prior work in BOBW literature has explored adaptive learning-rate schedules for FTPL. Nevertheless, we believe regret can be improved by adopting adaptive learning rates, such as arm-dependent learning rates, as successfully done for the FTRL framework as in Jin et al. (2023).
> The main idea behind arm-dependent is to balance the stability term and the penalty term so that they contribute to the regret in the same order. In our setting, this corresponds to making the first and second terms in (9) have the same order, which is related to Lemma 5 and Lemma 6.
>
> However, introducing arm-dependent learning rates does not allow a naive adaptation of Lemmas 5 and 6 by replacing $\eta_t$ with $\eta_{t,i}$ while keeping the similar definition of $q_{t,i}$ in (7). **Here, we want to clarify which part is not straightforward.**
>
> The original FTPL exploits an arm according to (a) $i\_t = \arg \min \\{\hat{L}\_{t,i} - r\_{t,i}/\eta\_t\\}$ whereas the current BOBW analysis is based on the formulation (b) $i\_t = \arg \min \\{ \eta\_t \hat{\underline{L}}\_{t,i} - r\_{t,i} \\}$.
> When the learning rate is identical for all arms, the two formulations coincide.
> However, with arm-dependent learning rates, (a) and (b) are no longer equivalent.
> Instead, the correct formulation becomes $i\_t = \arg \min \\{ \hat{\underline{L}}\_{t,i} - r\_{t,i}/\eta\_{t,i} \\}$, which corresponds to FTPL with arm-dependent perturbation distributions (and the common learning rate). In other words, designing arm-dependent learning rates is equivalent to designing FTPL with arm-dependent perturbations and arm-independent learning rates. This may require handling non-i.i.d. and potentially correlated perturbations, since $\underline{\hat{L}}_{t,i}$ depends on the trajectory of the current leader (the arm that minimizes $\hat{L}\_t$ at each round)
> Therefore, any attempt to mimic arm-dependent learning rate schemes within FTPL requires substantially more sophisticated analysis techniques, not only in our decoupled setting but even in standard MAB and related problems.
>
> In addition, our use of $q\_{t,i}$ as an approximation of $w\_{t,i}$ naturally suggests the possibility of arm-dependent learning rates, given that the learning rate $\eta\_{t,i}$ in Jin et al. (2023) is defined directly in terms of $w\_{t,i}$. Nevertheless, several nontrivial obstacles remain before such a design can be carried out in the FTPL framework.

---

### Official Review · Reviewer_VjFH · 2025-10-27

**Soundness:** 3
**Presentation:** 2
**Contribution:** 2
**Rating:** 4
**Confidence:** 3

**Summary:**

This paper investigates the decoupled bandit problem, a relatively less-studied topic within the bandit literature that possesses non-trivial application value. Specifically, the authors propose a policy based on the Follow-the-Perturbed-Leader (FTPL) framework using Pareto perturbations. The paper presents theoretical results for two cases: stochastic and adversarial environments. The new algorithm offers practical value, as it does not require solving optimization problems, unlike other methods. It also provides theoretical contributions, such as achieving a worst-case optimal regret bound in the adversarial setting. Experiments are provided to demonstrate the algorithm's effectiveness, though they utilize a limited choice of baselines.

**Strengths:**

1. The paper provides a comprehensive comparison with prior work and a sufficient background for the problem.
2. The algorithm demonstrates non-trivial improvements over existing methods.

**Weaknesses:**

1. The paper could benefit from adhering more strictly to writing conventions; notations are frequently used before they are defined (e.g., $w_t$ on line 136). It is strongly recommended to define mathematical notations before or as they are introduced in the same sentence to improve readability.
2. The analysis for the adversarial environment relies on a strong constraint, assuming that the gaps are constant.

**Questions:**

1. The authors comment that their problem is different from a pure exploration problem. Typically, an anytime pure exploration framework can be viewed as a process at each step involving: (1) selecting an arm, (2) pulling the arm and receiving a reward, and (3) recommending the arm currently inferred to be best. Note that the recommended arm (3) is not necessarily the same as the pulled arm (1), and it is only recommended, not pulled. I wonder what the precise difference is between this pure exploration setup and the problem addressed in this paper if we pull the arm in (3).

2. Could the authors compare the performance of their proposed algorithm with methods from the pure exploration, such as the Sequential Halving algorithm presented in "Revisiting simple regret: Fast rates for returning a good arm" (ICML 2023)? which is anytime.

3. Regarding the baseline EB-TC, how is the exploitation arm chosen? Is it the same as the arm that is sampled? This needs clarification.

4. It seems the confidence level for the experimental results are missing.

---

> ### Author Response · Authors · 2025-11-22
>
> Thank you for the time and effort you devoted to reviewing our paper. Below, we provide detailed responses to each of the weaknesses you raised.
>
> [Q1] As the reviewer pointed, any pure exploration policies can be directly applied in decoupled settings by exploiting currently best arm, i.e., exploitation with Follow-the-Leader (FTL). Therefore, the decoupled setting and pure exploration are closely related, but they differ fundamentally in **their objectives**. More precisely, the exploration in decoupled setting still implicitly considers the regret in exploitation while the exploration in pure exploration setting aims to reduce the uncertainty over the rewards. For this reason, while exponentially decreasing simple regret can be converted into small cumulative regret in stochastic decoupled setting, as shown Jourdan et al. (2023), the obtained regret bound becomes looser than policies for decoupled bandits.
>
> [Q2] We will incorporate the Sequential Halving (SH) algorithm into our experimental comparison. We are currently running the experiments, and we will provide the results as soon as the experiments finish.
>
> As shown in Figures 2 and 13 of Jourdan et al. (2023), SH exhibits inferior simple-regret performance compared to EB-TC. Based on this observation, we also expect SH to show lower cumulative-regret performance than EB-TC in the decoupled MAB setting.
>
> [Q3] As the reviewer mentioned in Q1, the exploited arm should be the recommended arm, that is, the currently best arm, as stated in Jourdan et al. (2023). However, we found that our previous code unintentionally used the sampled arm for exploitation. We have corrected this and will update all results including SH as soon as possible.
>
> [Q4] For the regret performance shown in Figure 1, we have uploaded PDF files at the anonymous link below, with shaded regions representing 95% and 99% confidence intervals, labeled “95 CI” and “99 CI,” respectively. If this is not what you intended by “confidence level,” please let us know and we will clarify further.
>
> Link to PDFs: https://anonymous.4open.science/r/anonymous-8A3C

---

> > ### Author Response · Authors · 2025-12-02
> >
> > We have now completed the additional experiments described above. The results and detailed explanations are provided in the PDF file “Additional experiments”, which we have uploaded to the anonymous link previously shared (https://anonymous.4open.science/r/anonymous-8A3C). These experiments include the corrected implementation of the exploited arm and the algorithm you suggested.
> >
> > Furthermore, we will incorporate these results into our paper to ensure that the experimental section more fully reflects the connection between pure exploration and the decoupled setting. We hope that these additions address the reviewer’s concerns and contribute to a clearer empirical understanding.

---

### Official Review · Reviewer_x6H3 · 2025-11-01

**Soundness:** 3
**Presentation:** 3
**Contribution:** 1
**Rating:** 2
**Confidence:** 4

**Summary:**

This paper revisits the decoupled bandit problem and proposes a computational efficient BOBW algorithm via FTPL, where the computational complexity improves by a $K$ factor.

**Strengths:**

1. The writing of this paper is clear and easy to follow.
2. The proposed algorithm enjoys an efficient computation complexity, while still achieving the best-of-both-world guarantee.

**Weaknesses:**

1. The primary concern of the reviewer is that the contribution and novelty of this paper are too thin to be accepted. Only one computational improvement is proposed, and the design and analysis of the FTPL technique largely follow prior works. The reviewer suggests that the authors consider extending their FTPL technique to a broader range of problems, especially more challenging setups, e.g., combinatorial bandits, rather than the simplified MAB setup in the current paper.
2. Several places in this paper are unclear:
    - The “$20$ times faster” claim is not clear.  Does that mean computational complexity, or just from the empirical experiments?
    - The “adaptive adversary” in Line 128 is an unclear term. It has two possibilities: strong (in response to current and past actions) and medium (only in response to past actions).
    - What is $\sigma_{t,i}$ in Eq.(7)?

**Questions:**

See weaknesses above.

---

> ### Author Response · Authors · 2025-11-22
>
> Thank you for the time and effort you devoted to reviewing our paper. Below, we provide detailed responses to the weaknesses you raised.
>
> [W1]
>
> **The importance of computational improvement**
>
> As we mentioned in the general response for our contribution (#1), we believe that improving computational efficiency is important, particularly in learning theory, where some theoretically grounded works still rely on policies that are computationally intractable. Moreover, our improvement is not about accelerating existing optimization approaches, but about **completely avoiding time-consuming optimizations and repeated resampling** while still preserving the BOBW guarantee. Since the original motivation of decoupled setting is related to communication engineering where the fast computation is important, we believe the contribution of computational improvement is not small.
>
> **Similarity to previous analysis**
>
> As we mentioned in the general response (#3), the similarity to previous analysis is due to the structural similarity to the standard MAB. Recall that the regret is determined by the performance of exploitation, and we exploit arms with the same policy used in MAB. This is also the case in the previous Decoupled-Tsallis-INF analysis, where the similar structure was used to the original BOBW analysis of Tsallis-INF.
>
> **Extension to the broader problems**
>
> First, we emphasize that the decoupled setting is **not a simplified MAB**. It strictly includes the standard MAB as a special case and can be roughly viewed as a MAB problem with side information. As we explained in our response to Reviewer tnbB (W1), the decoupled setting is valuable since the attainable regret bound is substantially different ($\log T$ regret to constant regret in stochastic setting).
>
> While we agree that extending BOBW results of FTPL is interesting, doing so requires addressing several nontrivial obstacles, as discussed in our response to Reviewer XB6E (Q1–Q2). The main obstacle is again related to the non-observed $w_t$. Recent advanced FTRL approaches explicitly use $w_t$ to design learning rates for difficult problems such as arm-dependent learning rate. Since $w_t$ of FTPL has no closed form, computing it directly introduces significant computational burden, which degrades the benefit of FTPL. For this reason, we believe our work provides a meaningful first step toward expanding the applicability of FTPL, independent of the immediate usefulness of the decoupled setting itself.
>
> [W2]
>
> (1) The phrase *"20 times faster"* refers to the **empirical runtime improvement, as reported in line 425.** We will make it clear that this phrase refers to the empirical computation time. Note that, to ensure a fair comparison, we implemented the FTRL-based methods using Newton's method to make them as efficient as possible. On the theoretical side, our proposed policy is at least $K^2/\log K$ (or $K$) times better than directly using GR (or CGR) in FTPL-based methods.
>
> (2) In our paper, the term *"adaptive adversary"* refers to the medium adaptive adversary that can **react to past actions, as defined in line 128.**
>
> (3) The definition of $\sigma_{t,i}$ **is indeed provided right after Eq.(7) in line 258**, where we explain that it denotes the rank of the cumulative loss.

---

### Author Response · Authors · 2025-11-22
**Detailed contribution statement**

We thank all reviewers for the time and effort they have invested in evaluating our submission. While we address the individual comments in detail below, we would like to clarify and highlight the core contributions of our work, which we feel are not fully reflected in the current reviews. Our goal in this rebuttal is to ensure that the key practical and theoretical advances of the paper are properly taken into account in the assessments.

Decoupled MABs are strongly motivated by latency-sensitive applications, such as communication engineering (Avner et al., 2012), where decisions must often be made within very short time scales (e.g., on the order of $1$ ms) and under noisy conditions. Hence, reducing computational overhead in decoupled MABs while maintaining BOBW guarantees is extremely essential. Addressing this challenge, we contribute substantially beyond simply removing the convex optimization step in FTRL via adapting FTPL: we further completely eliminate the need for geometric resampling (GR), a step long recognized as the **primary computational bottleneck** of FTPL (see #1 for details). This is a non-trivial contribution, as our new, efficiently computable sampling replacement allows FTPL to **fully realize its intended computational efficiency** (see #1 for details), serving as an efficient alternative to FTRL. Furthermore, as shown in Figure 1, our proposed method consistently outperforms the existing methods in decoupled MABs, achieving both practical superiority over previous BOBW algorithm and theoretical guarantees.

Beyond these practical gains, our work also provides meaningful theoretical advances. In particular, our construction of $q_t$ is **the first effective replacement for the $w_t$** used in recent BOBW FTRL policies, enabling this component to be incorporated within an FTPL framework (see #2 for details). As a direction for future work, this approximation technique can be utilized for designing adaptive learning rates for FTPL, an area where progress has been relatively stalled due to the absence of closed-form for arm-selection probabilities.

---

> ### Author Response · Authors · 2025-11-22
>
> 1. **GR has been a practical bottleneck of FTPL in standard bandits.**
>
> Since FTPL leverages randomness by adding perturbations, it can select arms with a similar probability to FTRL policy, without explicitly computing $w_t$ via convex optimization problem. For this property, FTPL is generally considered as a computationally efficient alternative to FTRL in online learning literature. However, conversely, this property becomes problematic when the explicit value of $w_t$ is required. This is usually the case when we construct an unbiased loss estimator in bandit feedback,$^1$ which makes the practical efficiency of FTPL be relatively limited in standard bandit settings.
>
> In standard bandit settings, FTPL often relies on GR procedure to estimate $w_{t,{i_t}}^{-1}$ for the importance-weighted (IW) loss estimator instead of computing full vector $w_t$. However, GR incurs a per-round average complexity of $\mathcal{O}(K^2)$, which is usually worse than that of FTRL for solving convex optimization problem in the standard MAB setting.$^2$ More specifically, when the solution of convex optimization is of the closed form, such as Tsallis-INF, the use of Newton's method is actually more computationally efficient than FTPL with GR. This has been widely recognized as the main obstacle to FTPL's broader applicability, motivating recent work that reduces GR's complexity to $\mathcal{O}(K \log K)$ (Chen et al. 2025).
>
> Unlike the standard MAB, the decoupled setting allows the exploration strategy $p_t$ and the exploitation strategy $w_t$ to be designed separately. This means that the explicit computation of $w_t$ may not be required if we set $p_t$ independent to the value of $w_t$. However, the currently known policies set $p_t$ in terms of $w_t$: Exp3-type (Avner et al., 2012) and Decoupled-Tsallis-INF (Rouyer & Seldin 2021). While Exp3-type is computationally efficient, it does not have BOBW guarantee. On the other hand, though Decoupled-Tsallis-INF achieves BOBW, it requires solving optimization problem to compute $w_t$. Given that one of the original motivations for the decoupled setting is the applications in communication engineering (Avner et al., 2012), where decisions often need to be made in under $1$ms and the environment often be noisy, fast per-iteration computation with BOBW guarantee is critical. In this sense, the use of FTPL would be promising considering the recent advances of FTPL in BOBW literature. However, the straightforward adaptation of $p_t$ with $w_t$ requires to compute the full vector, which makes the use of FTPL meaningless. In this paper, we entirely eliminate the need for GR by introducing a new sampling replacement, which enables FTPL to fully realize its **intended efficiency advantage**. We believe this improvement is non-trivial as it preserves the theoretical guarantees of FTPL while significantly reducing its computational overhead.
>
> - 1) While FTRL with reduced variance (RV) estimator showed the better empirical performance than that with importance-weighted estimator (Zimmert & Seldin, 2021), the design of RV estimator for FTPL is not straightforward as its definition includes the value of $w_t$.
>
> - 2) GR was originally proposed for combinatorial bandits (Neu & Bartok, 2016), where it is far more efficient than solving the convex optimization problem in FTRL, as the underlying convex set is much larger than in the standard MAB setting.

---

> ### Author Response · Authors · 2025-11-22
>
> 2. **Efficient construction of the exploration probability**
>
> The core contribution of our work lies in the policy design itself, most notably in the construction of the exploration probabilities $p_t$. By introducing an efficiently computable approximation $q_t$ instead of directly estimating $w_t$, we resolve the computational bottleneck and opens new directions for FTPL research on adaptive learning rates. This is particularly important because recent state-of-the-art (SOTA) FTRL approaches in the BOBW literature rely explicitly on $w_t$, including applications such as graph feedback and partial monitoring (Tsuchiya & Ito, 2024). By analogy, we expect that in FTPL, our approximation technique for $w$ can provide a foundation for the principled design of learning rates and potentially enable FTPL to obtain stronger guarantees and broader applicability.
>
> Since recent advanced FTRL methods often employ hybrid regularizers, Newton’s method is not even applicable, which makes a computationally practical alternative necessary. Therefore, our construction $q_t$ for FTPL, replacing the dependence on $w_t$ that FTRL requires, provides a clear conceptual and technical contribution and opens a path for deploying FTPL in settings far beyond the decoupled case.
>
> [1] Taira Tsuchiya and Shinji Ito, A Simple and Adaptive Learning Rate for FTRL in Online Learning with Minimax Regret of $\Theta(T^{2/3})$ and its Application to Best-of-Both-Worlds, NeurIPS 2024.
>
> 3. **Similarity to prior analysis**
>
> Although one might argue that our analysis follows earlier BOBW analyses of FTPL, such similarities are natural and unavoidable due to the structural similarity between standard MABs and decoupled MABs since we exploit arms using the same policy as in the standard MAB. The same situation is observed in the previous decoupled policy of Rouyer & Seldin (2020), whose analysis also closely follows that of Zimmert & Seldin (2021) for FTRL with Tsallis entropy.

---

### Note · Authors · 2026-01-26

**Comment:**

We thank the reviewers and the AC for the time and effort they invested in reviewing our submission. With all due respect, we unfortunately believe that the current review comments do not fully reflect the significance of our results and contributions, possibly due to shortcomings in our presentation. We will revise the manuscript to make the key ideas and findings clearer and more fully supported, and we hope to resubmit an improved version in the future.

**Withdrawal Confirmation:**

I have read and agree with the venue's withdrawal policy on behalf of myself and my co-authors.

---

### Meta-Review · Area_Chair_kCrM · 2026-01-04

**Summary:**

The reviewers raised the following key concerns. First, most reviewers expressed concerns regarding the significance of the results and limited technical contributions of the current submission. Second, reviewer XB6E pointed out that the regret bound in the stochastic setting is worse than existing results by Jin et al. (2023). Third, reviewer VjFH noted that results in the adversarial setting rely on relatively strong assumptions. Finally, there were concerns regarding clarity and paper presentation.

**Reviewer Concerns:**

The rebuttal addressed concerns on paper presentations and clarified several questions raised by the reviewers. The rebuttal also partially addressed the first concern by emphasizing the importance of the studied problem and their technical contributions on improved computational efficiency. The authors acknowledged that their algorithm can be $\sqrt{K}$ worse than the approach of Jin et al. (2023) in terms of regret bound, but argued that it offers improved computational efficiency. The rebuttal didn't respond to the concern regarding the strong assumptions in the adversarial setting, as raised by reviewer VjFH.

**Reviewer Scores:**

If the reviewers had been able to participate fully in the discussion, I believe reviewer x6H3 might have increased their score from 2 to 4, while the other reviewers' scores would have remained unchanged at 4.

---

### Decision · Program_Chairs · 2026-01-26

Reject